# HDAC6-dependent deacetylation of SAE2 enhances SUMO1 conjugation for mitotic integrity

Alexander J Lanz[1,4], Alexandra K Walker[1,4], Mohammed Jamshad[1], Alexander J Garvin [ID][1,2],
Matthew Stewart[1], Peter Wotherspoon [ID][3], Benjamin F Cooper [ID][3], Matthew Mackintosh[1],
Oliver Crutchley [ID][1], Timothy J Knowles [ID][3] & Joanna R Morris [ID][1✉]

## Abstract

**Mammalian cells express three conjugatable SUMO variants: SUMO1 and the closely related SUMO2 and SUMO3 (together referred to as SUMO2/3). While some substrates are modified by both, others show a clear preference, though the basis for this selectivity remains unclear. Here, we examine a modification of the catalytic component of the human SUMO activation enzyme, SAE2. We find that lysine 164 of SAE2 undergoes HDAC6-dependent deacetylation during mitosis. A non-deacetylatable acetyl-mimetic mutant, SAE2-K164Q, selectively enhances SUMO2 over SUMO1 activation and conjugation, and distinguishes between SUMO1 and SUMO2/3 based on differences in their C-terminal tails. Complementation of SAE2-deficient or inhibited cells with SAE2-K164Q suppresses mitotic SUMO1 conjugation and promotes multipolar spindle formation. We identify NuMA as a SUMO E1-dependent substrate and demonstrate that mitotic defects caused by SAE2-K164Q or HDAC6 inhibition are rescued by SUMO1 overexpression or expression of a GFP-SUMO1-NuMA-K1766R fusion. These results support a model in which SAE1:SAE2 deacetylation during early mitosis promotes SUMO1 conjugation to ensure mitotic fidelity, highlighting a regulatory role for the SUMO-activating enzyme in the selection of SUMO proteins.**

**Keywords** SUMOylation; Mitosis; E1; HDAC; Acetylation
**Subject Categories** Cell Cycle; Post-translational Modifications & Proteolysis

## Introduction

Mammals have ~20 ubiquitin-like modifiers, including the Small Ubiquitin-like Modifiers (SUMOs) (Ilic et al, 2022). Higher eukaryotes express two subfamilies of conjugatable SUMO variants: SUMO1 and the highly similar SUMO2 and SUMO3, referred to as SUMO2/3. SUMOylation contributes to many intracellular processes, including transcription, DNA repair, chromatin remodelling, signal transduction, and mitosis (Chang and Yeh, 2020; Garvin and Morris, 2017). In the initial step of the SUMOylation cascade, SUMO activation is catalysed by the sole SUMO E1 heterodimer comprising SUMO-Activating Enzyme 1 (SAE1/AOS1) and 2 (SAE2/UBA2) (Desterro et al, 1999; Johnson et al, 1997; Okuma et al, 1999); herein referred to as SAE1:SAE2. The SAE1:SAE2 adenylation catalytic site coordinates SUMO and ATP-Mg$^{2+}$ and initiates the conjugation process by adenylating the SUMO C-terminus, producing SUMO-AMP. Subsequently, SAE1:SAE2 undergoes remodelling around SUMO-AMP to form the thiolation catalytic site for thioester bond formation between the SUMO C-terminus and SAE2-C173 (Lois and Lima, 2005). The activated SUMO is then transferred to the catalytic cysteine in the only SUMO-conjugating enzyme, UBC9 (Johnson and Blobel, 1997). SUMO can then be conjugated onto a target lysine directly from UBC9 with or without the added guidance of a SUMO E3 ligase.

Thousands of proteins are SUMOylated and deSUMOylated in a spatially and temporally controlled manner (Becker et al, 2013; Hendriks et al, 2018; Hendriks et al, 2017; Hendriks and Vertegaal, 2016a). SUMO1 is conjugated to its substrates chiefly as a single conjugate (mono-SUMOylation), with a small proportion in SUMO chains, whereas SUMO2/3 more often form chains (poly-SUMOylation) (Hendriks and Vertegaal, 2016b, Ulrich, 2008). Most SUMO1 in mammalian cells appears conjugated to proteins, whereas much of SUMO2/3 is found unconjugated, and their conjugation to substrates is increased following cellular stresses (Ilic et al, 2022; Saitoh and Hinchey, 2000). Some substrates are modified by SUMO1, others by SUMO2/3, and many by both forms (Becker et al, 2013). Three broad mechanisms promote differences in substrate-SUMO variant modifications; SUMO-variant-specific SIMs (SUMO interaction motifs) in SUMO E3s and target proteins can bias modification (Chang et al, 2009; Gareau et al, 2012; Hecker et al, 2006; Meulmeester et al, 2008; Namanja et al, 2012; Reverter and Lima, 2005; Tatham et al, 2005; Zhu et al, 2008); a SUMOylated variant may be protected from isopeptidases after non-specific modification (Werner et al, 2012; Zhu et al, 2009) or

[1]Birmingham Centre for Genome Biology and Department of Cancer and Genomic Sciences, School of Medical Sciences, College of Medicine and Health, University of Birmingham, Birmingham B15 2TT, UK. [2]SUMO Biology Lab, School of Molecular and Cellular Biology and Astbury Centre for Structural Molecular Biology, Faculty of Biological Sciences, University of Leeds, Leeds LS2 9JT, UK. [3]School of Biosciences, University of Birmingham, Birmingham B15 2TT, UK. [4]These authors contributed equally: Alexander J Lanz, Alexandra K Walker. ✉E-mail: j.morris.3@bham.ac.uk

*The EMBO Journal* Volume 44 | Issue 19 | October 2025 | 5537 – 5563 **5537**

SUMO variant and conjugate types may be differentially processed by SUMO proteases (SENPs), some of which exhibit bias.

Accurate mitotic signalling is essential for faithful chromosome segregation and the generation of genetically stable daughter cells. The SUMO pathway plays a critical role in this process. Depletion or inhibition of SAE1:SAE2, or loss, depletion, or mutation of UBC9 causes delays in chromosome alignment, errors in segregation, and impaired metaphase-to-anaphase transitions (Azuma et al, 2003; Eifler et al, 2018; He et al, 2017; Nacerddine et al, 2005 and reviewed in Abrieu and Liakopoulos, 2019). SUMO1 and SUMO2/3 appear to have differing roles in this portion of the cell cycle. Over 70 SUMO2/3 mitotic substrates have been described (Merbl et al, 2013; Schimmel et al, 2014; Schou et al, 2014). SUMO2/3 localises to mitotic DNA, and SUMO2/3ylated PARP1 (Ryu et al, 2010) and TOP2 (Azuma et al, 2003) are present on mitotic chromatin. SUMO2/3 is also found at the protein complexes that attach chromosomes to spindle microtubules, the kinetochores, where SUMOylation promotes the recruitment of many proteins, including PLK1 (Feitosa et al, 2018), PICH (Sridharan and Azuma, 2016) and Aurora B (Ban et al, 2011; Fernandez-Miranda et al, 2010). SUMO1 appears on the structures that help organise microtubules, the centrosomes, and also along the microtubules of the mitotic spindle (Zhang et al, 2008). The majority of the observed SUMO1-localisation is RanGAP1-SUMO1 (Joseph et al, 2002; Zhang et al, 2008) and few other mitotic SUMO1 substrates have been reported; they include BubR1 (Yang et al, 2012), Aurora-A (Perez de Castro et al, 2011), PLK1 (Feitosa et al, 2018; Wen et al, 2017) and NuMA (Seo et al, 2014). Defective SUMO1ylation of these substrates is associated with prolonged mitosis (Wen et al, 2017) or spindle abnormalities (Joseph et al, 2002; Seo et al, 2014). The mechanism by which the SUMO conjugation machinery achieves SUMO variant-specific modification of target proteins, particularly during mitosis, remains unclear.

Here, we identify a previously undescribed means of SUMO variant-conjugation bias. We find that SAE2 is deacetylated during mitosis and that deacetylation can be prevented with HDAC6 inhibitor treatment. We find that a SAE2 acetyl-analogue, SAE2-K164Q, drives a SUMO2 > SUMO1 conjugation bias, resulting in diminished high-molecular-weight SUMO1 conjugates in mitotic cells. Cells complemented with SAE2-K164Q exhibit supernumerary structures of the nuclear mitotic apparatus NuMA, multipolar spindles, and CENPA-positive micronuclei indicative of poor chromosome segregation. Remarkably, these mitotic defects, whether caused by SAE2-K164Q expression or HDAC6 inhibition, are rescued by SUMO1 overexpression or expression of a SUMO1-NuMA, but not SUMO2-NuMA fusion protein, implicating insufficient SUMO1ylation of NuMA as the primary cause of spindle defects. Together, we identify a previously unrecognised mechanism driving SUMO conjugation toward specific variants, revealing that bias can originate at the level of the SUMO E1 enzyme and directly impact mitotic fidelity.

# Results

## Acetylated-K164-SAE2 is deacetylated by HDAC6

Acetylation of SAE2 at K164, acK164-SAE2, was identified in HeLa cells by liquid chromatography–mass spectrometry, following peptide separation by strong cation exchange chromatography and enrichment with a pan-acetyl antibody (Elia et al, 2015). To study this modification, we generated a monoclonal antibody specific to the K164-acetylated SAE2 peptide, "HP[ac]KPTQRTFPGC". We also established U2OS cell lines with Doxycycline-inducible expression of either wild-type (WT) FLAG-SAE2 or a FLAG-SAE2-K164R mutant. Testing the antibody on immunoprecipitated proteins revealed that it robustly detected WT FLAG-SAE2 but not the K164R mutant (Fig. EV1A).

Previous mass spectrometry experiments suggested that acetylation at K164 of SAE2 may be lost following irradiation (Elia et al, 2015). To test this, we examined precipitated WT FLAG-SAE2 for the acK164-SAE2 signal an hour after irradiation exposure (10 Gy), confirming a reduction of acK164-SAE2, with no change detected in total FLAG-SAE2 protein after treatment (Fig. EV1A). Further, we noted reduced acK164-SAE2 signal in lysates from cells treated with the microtubule polymerisation inhibitor nocodazole and harvested by mitotic shake-off (Fig. 1A), suggesting that deacetylation coincides with early mitosis.

We tested several inhibitors of deacetylases and noted that the inhibitor Panobinostat, a broad-spectrum Histone Deacetylase (HDAC) inhibitor targeting Class I, II, and IV, prevented the loss of the acK164-SAE2 signal. Similar effects were observed with ACY-738, a selective HDAC6 inhibitor (Jochems et al, 2014) (Fig. 1A). To explore whether HDAC6 interacts with SAE2, we performed co-immunoprecipitation assays. Endogenous HDAC6 co-precipitated with FLAG-SAE2, and this interaction was further enriched in mitotic cells synchronised using nocodazole and harvested by mitotic shake-off (Fig. 1B). These data suggest SAE2-K164 deacetylation occurs through the interaction and activity of HDAC6 and is increased in early mitosis.

Next, we addressed whether specific acetylase enzymes contribute to the modification. We tested inhibitors of p300, TIP60, NAT10, and GCN5 and found that no single inhibitor application suppressed the detection of acK164-SAE2. However, combining NAT10, TIP60, and p300 inhibitors reduced the acK164-SAE2 signal, albeit not completely (Fig. 1C). These data suggest that SAE2-K164 acetylation occurs via the activity of more than one acetylase enzyme.

## SAE2-K164Q has a SUMO2 > SUMO1 bias

Residue K164 of SAE2 forms part of the tunnel through which the C-terminal tails of SUMO proteins access the SAE1:SAE2 catalytic sites in SUMO activation (Hann et al, 2019; Lois and Lima, 2005; Olsen et al, 2010) (Fig. EV1B). This proximity led us to query whether K164-SAE2 modification might impact SUMO protein interactions. We generated recombinant (WT) SAE1:SAE2 proteins and a mutant in which the SAE2 element carried a mutation at residue 164 to glutamine (Q) (Fig. EV2: Expanded View Fig. 2 contains SDS-PAGE gels of SEC fractions of all the purified proteins generated in the current study). Glutamine resembles the uncharged carbonyl functional group of an acetylated lysine, and although not forming a classical isostere, glutamine has been used as an acetyl-analogue (Bhardwaj and Das, 2016; Gartner et al, 2018; Kamieniarz and Schneider, 2009; Kim et al, 2006). We labelled SAE1:SAE2 and SAE1:SAE2-K164Q with NT-647-NHS for Micro-Scale Thermophoresis (MST). This technique detects the movement of fluorophore-tagged molecules in a temperature gradient.

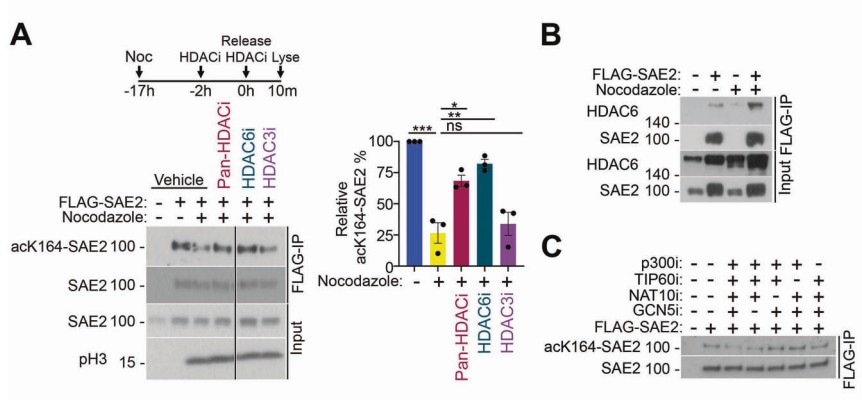

**Figure 1. Acetylation of SAE2-K164.**

(A) Western blot, representative of three, of acetyl-K164-SAE2 (mouse monoclonal) following anti-FLAG immunoprecipitation from U2OS cells and U2OS cells expressing FLAG-SAE2 in unsynchronised cells or cells treated with nocodazole for 17 h before washing and releasing into mitosis for 10 min. HDAC inhibitors, Panobinostat (HDAC class I, II, and IV), ACY-738 (HDAC6), and RGFP966 (HDAC3), were applied to cells in the last 2 h of nocodazole synchronisation at 2.5 μM and reapplied upon the release. An antibody specific for phosphorylated-Ser10 on histone 3 is used as a marker of mitosis. Quantification of the relative abundance of acetyl-K164-SAE2 relative to the total abundance of SAE2 immunoprecipitated. Error bars = SEM; N = 3 biological repeats. ***P < 0.001, **P < 0.01, *P < 0.05, ns = not significant P > 0.05. Vehicle vs Nocodazole P = 0.0002, Nocodazole vs Pan-HDACi P = 0.0143, Nocodazole vs HDAC6i P = 0.0019, Nocodazole vs HDAC3i P = 0.9494. Statistical significance was calculated using one-way ANOVA. (B) Western blot analysis of a FLAG-SAE2 co-immunoprecipitation with HDAC6 from U2OS, in the context of an asynchronous cell population or following a 16 h nocodazole-treatment and mitotic shake-off. Replicated twice in the laboratory. (C) Western blot of acK164-SAE2 following anti-FLAG immunoprecipitation from U2OS cells and U2OS cells expressing FLAG-SAE2 in asynchronous cells treated with indicated combinations of inhibitors against the histone acetyltransferases p300 (A-485), TIP60 (NU9056), NAT10 (Remodelin Hydrobromide), and GCN5 (Butyrolactone). Inhibitors were added for 2 h at 2.5 μM. Replicated three times in the laboratory.

The thermophoresis of a protein differs from that of its liganded complex, such that MST can be used to quantify interaction dissociation constants (Wienken et al, 2010). We subjected labelled SAE1:SAE2 to MST analysis with increasing concentrations of SUMO1 or SUMO2. Fitting of the data to the change in thermophoresis showed that SAE1:SAE2 has a greater affinity (lower $K_d$) for SUMO1 (3.7 ± 1.1 μM) than SUMO2 (14.7 ± 1.8 μM; Figs. 2A and EV3A), consistent with $K_m$ measurements of SUMO1 vs SUMO2 with SAE1:SAE2 (Wiryawan et al, 2015). Intriguingly, performing the same analysis with SAE1:SAE2-K164Q revealed a greater affinity (lower $K_d$) for SUMO2 (0.4 ± 0.13 μM) and reduced affinity for SUMO1 (28.0 ± 12.19 μM; Figs. 2A and EV3A). These data show that residue 164 of SAE2 can influence the affinity of SAE1:SAE2 for SUMO proteins.

To assess the SUMO activation capacity of SAE1:SAE2 vs SAE1:SAE2-K164Q, we generated two additional recombinant SAE1:SAE2 variants, SAE1:SAE2-K164R, and a thiolation catalysis mutant SAE1:SAE2-C173G. We tested SUMO protein adenylation in vitro through incubation of SAE1:SAE2 variants with SUMO1 or SUMO2 and fluorescently labelled ATP, BODIPY-ATP, followed by the quantification of SUMO1-AMP-BODIPY or SUMO2-AMP-BODIPY (Olsen et al, 2010). We first tested the SAE1:SAE2-C173G mutant form of the enzyme. The C173G mutant can catalyse SUMO-AMP formation but lacks the catalytic cysteine needed for thioester bond formation, and incubation with SAE1:SAE2-C173G resulted in high SUMO1- and SUMO2-AMP-BODIPY levels (Figs. 2B and EV3B). In contrast, incubation with WT SAE1:SAE2 enzyme produced low levels of SUMO-AMP-BODIPY (Figs. 2B and EV3B). By comparison, SAE1:SAE2-K164Q showed elevated SUMO1 adenylation, while the levels of SUMO2 adenylation were

comparable to those of the WT enzyme (Figs. 2B and EV3B). Incubation with the structurally similar SAE1:SAE2-K164R mutant resulted in SUMO1-AMP-BODIPY and SUMO2-AMP-BODIPY to a comparable extent as observed for WT SAE1:SAE2 reactions.

To test SAE2~SUMO thioester formation, we first generated fluorescent SUMO proteins, SUMO1-C52A-S9C-Alexa488 and SUMO2-C48A-A2C-Alexa647 (Alegre and Reverter, 2011; Cappadocia et al, 2015). We confirmed their ability to form DTT-sensitive SAE2~SUMO thioester in non-reducing gels (Fig. EV3C). To test the relative SAE2~SUMO thioester formation, we combined recombinant SAE1:SAE2-K164 variants with these SUMOs, ATP buffer, and examined products using SDS-PAGE under non-reducing conditions, monitoring the formation of the 120 kDa fluorescent SUMO band consistent with loaded SAE2~SUMO (Figs. 2C and EV3D). Interestingly, the reaction involving SAE1:SAE2-K164Q with SUMO1 showed a reduction in SAE2~-SUMO1 product. In contrast, the reaction between SAE1:SAE2-K164Q and SUMO2 did not significantly deviate from the WT SAE1:SAE2 reaction (Figs. 2C and EV3D). These findings imply that the SAE1:SAE2-K164Q mutant can generate adenylated SUMO1, but its ability to convert SUMO1-AMP to the thioester-linked SAE2~SUMO1 is inefficient. In contrast, the ability of SAE1:SAE2-K164Q to process SUMO2 is unaffected.

Next we wished to test whether SAE2-K164 influences substrate SUMOylation, particularly when both SUMO1 and SUMO2 are available. We used SUMO1 and SUMO2, UBC9, and a model substrate; the RanGAP1 fragment (amino acids 398–587). We found that all SAE1:SAE2 enzyme variants tested could promote RanGAP1 SUMOylation to a comparable degree after a 10-min reaction when supplied with single SUMO variants (Figs. 2D and EV3E,F). However,

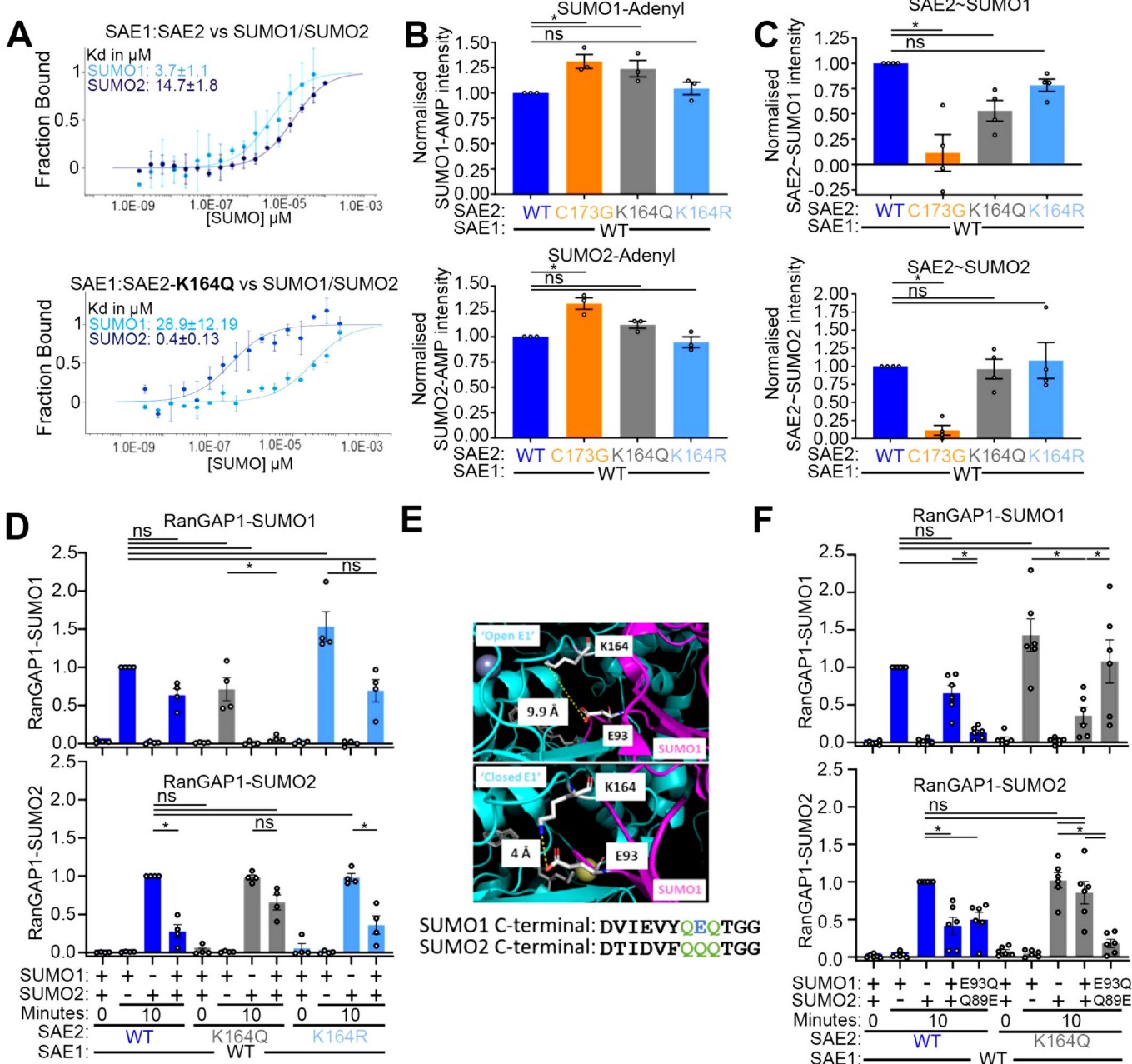

when incubated with equimolar amounts of SUMO1 and SUMO2, conjugates driven by SAE1:SAE2-K164Q exhibited a variant bias that differed from that driven by WT SAE1:SAE2 or SAE1:SAE2-K164R. If no bias were present, we would expect a suppression of 50% in SUMO1 and SUMO2 conjugation by the equimolar introduction of the other SUMO variant, relative to when each variant was supplied alone. For WT SAE1:SAE2 and SAE1:SAE2-K164R containing reactions, SUMO1 conjugates were reduced to 0.63 ± 0.09 and 0.69 ± 0.15, respectively, by equimolar SUMO2 (Figs. 2D and EV3E,F). SUMO2 conjugates were reduced to 0.27 ± 0.09 and 0.36 ± 0.12, respectively, by equimolar SUMO1 (Figs. 2D and EV3E,F). Thus, reactions containing WT-SAE1:SAE2 and SAE1:SAE2-K164R exhibit a slight SUMO1 > SUMO2 bias in the presence of both SUMO

variants. In contrast, in SAE1:SAE2-K164Q:SUMO1:SUMO2 containing reactions, SUMO1 conjugates were reduced to 0.06 ± 0.02, relative to the SUMO1-only reaction, by equimolar SUMO2, and SUMO2 conjugates were reduced to 0.66 ± 0.10, relative to the SUMO2-only reaction, by equimolar SUMO1 (Figs. 2D and EV3E,F). These data show that SAE1:SAE2-K164Q supports SUMO2 > SUMO1 activation, with SUMO1 conjugates almost eliminated in the presence of SUMO2. These findings are also consistent with the notion that SUMO1:-SAE1:SAE2-K164Q interaction is retained, since SUMO1 presence suppresses SUMO2 conjugate production, even though SUMO1 itself is poorly conjugated.

To address how residue 164 of SAE2 discriminates between SUMO proteins, we examined published structures of SAE1:SAE2-

**Figure 2. SUMO activation by SAE1:SAE2-K164Q is biased towards SUMO2.**

(A) MST analysis of SAE1:SAE2-K164Q affinity with SUMO1 and SUMO2. SUMO1 and SUMO2 concentrations were titrated from 125 μM to 0.00381 μM against 10 nM of NT647-labelled, SAE1:SAE2 (top) or SAE1:SAE2-K164Q (bottom). Plotting of the change in thermophoresis and fitting of the data yielded a $K_d$ in μM shown. $N = 3$ independent technical repeats, error bars = SD about the mean. (B) In vitro SUMO1 adenylation assay, conducted by combining 30 μM SAE1:SAE2-K164 variants with 40 μM SUMO1 or SUMO2, 150 μM BODIPY-ATP, and 1 U pyrophosphatase for 10 min at 30 °C. Reactions were quenched by the addition of loading buffer and incubation at 95 °C for 5 min before running samples on 15% SDS-PAGE. Gels were imaged at 488 nm to excite BODIPY-ATP, where a band at 15 kDa was taken to be SUMO-AMP-BODIPY (Fig. EV3B shows a representative gel). $N = 3$ technical repeats, error bars = SEM, and statistical significance calculated using one-way ANOVA where *$P < 0.05$. Statistical values for SAE2-SUMO1 intensity SAE2-WT vs SAE2-C173G $P = 0.0079$, SAE2-WT vs SAE2-K164Q $P = 0.0261$, SAE2-WT vs SAE2-K164R $P = 0.8328$. Statistics for SAE2-SUMO2 SAE2-WT vs SAE2-C173G $P = 0.0017$, SAE2-WT vs SAE2-K164Q $P = 0.1976$, SAE2-WT vs SAE2-K164R $P = 0.7038$. (C) In vitro SUMO loading assays combined 5 μM SAE1:SAE2-K164 variants or SAE1:SAE2-C173G with 5 mM ATP. Reactions were initiated by the addition of 10 μM SUMO on ice for 15 s and terminated by the addition of reducing agent-free loading buffer and boiling samples. Samples were analysed by SDS-PAGE where fluorescent SUMO bands at 120 kDa were taken to be SAE2-SUMO, which were quantified and normalised to the WT SAE1:SAE2 condition. $N = 4$ technical repeats, error bars = SEM, with statistical significance calculated using one-way ANOVA, *$P < 0.05$. Statistical values for SAE2-SUMO1 SAE-WT vs SAE-C173G $P = 0.0351$, SAE-WT vs SAE-K164Q $P = 0.0405$, SAE-WT vs SAE-K164R $P = 0.0765$. Statistics for SAE2-SUMO2 SAE-WT vs SAE-C173G $P = 0.0028$, SAE-WT vs SAE-K164Q $P = 0.9951$, SAE-WT vs SAE-K164R $P = 0.9676$. (D) In vitro SUMOylation of RanGAP1 (aa 398–587) fragment in the presence of 10 μM SUMO1 and/or 10 μM SUMO2, 25 nM SAE1:SAE2, 100 nM UBC9, 10 μM RanGAP1 (aa 398–587), and 5 mM ATP, incubated at 30 °C for 10 min. Fig. EV3E shows representative SUMOylation assays. The RanGAP1-SUMO1 products are normalised to the WT SAE1:SAE2:SUMO1-only condition (top), and the RanGAP1-SUMO2 products are normalised to the WT SAE1:SAE2:SUMO2-only condition (bottom). $N = 4$ technical repeats, error bars = SEM, with statistical significance calculated using one-way ANOVA, *$P < 0.05$. SUMO1 blot statistical values for SAE2-WT + SUMO1 vs SAE2-WT + SUMO1 + SUMO2 $P = 0.093$, SAE2-WT + SUMO1 vs SAE2-K164Q + SUMO1 $P = 0.4878$, SAE2-WT + SUMO1 vs SAE2-K164R + SUMO1 $P = 0.2668$, SAE2-WT + SUMO1 vs SAE2-K164R + SUMO1 + SUMO2 $P = 0.4222$, SAE2-WT + SUMO1 vs SAE2-K164Q + SUMO1 + SUMO2 $P < 0.0001$, SAE2-K164Q + SUMO1 vs SAE2-K164Q + SUMO1 + SUMO2 $P = 0.0381$, SAE2-K164R + SUMO1 vs SAE2-K164R + SUMO1 + SUMO2 $P = 0.0171$. SUMO2 blot statistical values for SAE2-WT + SUMO2 vs SAE2-WT + SUMO1 + SUMO2 $P = 0.0162$, SAE2-WT + SUMO2 vs SAE2-K164Q + SUMO2 $P = 0.9953$, SAE2-WT + SUMO2 vs SAE2-K164R + SUMO2 $P > 0.9999$, SAE2-K164Q + SUMO2 vs SAE2-K164Q + SUMO1 + SUMO2 $P = 0.0605$, SAE2-K164R + SUMO2 vs SAE2-K164R + SUMO1 + SUMO2 $P = 0.0213$. (E) Examination of human SUMO E1 K164-SAE2 (Cyan) in 'open' conformation (top) with E93-SUMO1-AMSN (magenta; PDB ID code 3KYC) and in the 'closed' conformation (bottom) with E93-SUMO1-AVSN (magenta; PDB ID code 3KYD) (Olsen et al, 2010). In the 'open' conformation (top) the SAE2-K164 ζ-nitrogen and SUMO1-E93 ε-carbonyl are 9.9 Å apart, while the 'closed' SUMO E1 conformation shows SAE2-K164 ζ-nitrogen and SUMO1-E93 ε-carbonyl ~4 Å apart (Olsen et al, 2010), a suitable distance for a noncovalent interaction. Amino acid sequence alignment for SUMO1-E93 and SUMO2-Q89. (F) In vitro SUMOylation RanGAP1 (aa 398–587) in the presence of WT SUMO1, and/or SUMO2, SUMO1-E93Q and SUMO2-Q89E (all SUMO variants supplied at 10 μM), 25 nM SAE1:SAE2, 100 nM UBC9, 10 μM RanGAP1 (aa 398–587), and 5 mM ATP, incubated at 30 °C for 10 min. Fig. EV3F shows representative SUMOylation assays. The RanGAP1-SUMO1 products are normalised to the WT SAE1:SAE2:SUMO1-only condition (top) and the RanGAP1-SUMO2 products are normalised to the WT SAE1:SAE2:SUMO2-only condition (bottom). $N = 4$ technical repeats, error bars = SEM, *$P < 0.05$. Statistical significance was calculated using one-way ANOVA. SUMO1 blot statistical values for SAE2-WT + SUMO1 vs SAE2-WT + SUMO1 + SUMO2 $P = 0.0914$, SAE2-WT + SUMO1 vs SAE2-WT + SUMO1-E93Q + SUMO2-Q89E $P < 0.0001$, SAE2-WT + SUMO1 vs SAE2-K164Q + SUMO1 $P = 0.3888$, SAE2-WT + SUMO1 vs SAE2-K164Q + SUMO1 + SUMO2 $P = 0.0114$, SAE2-WT + SUMO1 vs SAE2-K164Q + SUMO1-E93Q + SUMO2-Q89E $P > 0.9999$, SAE2-WT + SUMO1 + SUMO2 vs SAE2-WT + SUMO1-E93Q + SUMO2-Q89E $P = 0.0029$, SAE2-K164Q + SUMO1 vs SAE2-K164Q + SUMO1 + SUMO2 $P = 0.0038$, SAE2-K164Q + SUMO1 + SUMO2 vs SAE2-K164Q + SUMO1-E93Q + SUMO2-Q89E $P = 0.0445$. SUMO2 blot statistical values for SAE2-WT + SUMO2 vs SAE2-WT + SUMO1 + SUMO2 $P = 0.0186$, SAE2-WT + SUMO2 vs SAE2-WT + SUMO1-E93Q + SUMO2-Q89E $P = 0.0185$, SAE2-WT + SUMO2 vs SAE2-K164Q + SUMO2 $p > 0.9999$, SAE2-WT + SUMO2 vs SAE2-K164Q + SUMO1 + SUMO2 $P = 0.8984$, SAE2-WT + SUMO1 + SUMO2 vs SAE2-WT + SUMO1-E93Q + SUMO2-Q89E $P = 0.4688$, SAE2-K164Q + SUMO2 vs SAE2-K164Q + SUMO1 + SUMO2 $P = 0.2231$, SAE2-K164Q + SUMO2 vs SAE2-K164Q + SUMO1-E93Q + SUMO2-Q89E $P = 0.0003$, SAE2-K164Q + SUMO1 + SUMO2 vs SAE2-K164Q + SUMO1-E93Q + SUMO2-Q89E $P = 0.0057$.

SUMO1-AVSN (Hann et al, 2019; Olsen et al, 2010) and noted proximity between lysine 164 of SAE2 and glutamate 93 of SUMO1 (Fig. 2E). Intriguingly, the equivalent residue in SUMO2 is glutamine (Fig. 2E). To test the hypothesis that SAE2 residue 164 discriminates SUMO proteins through these residues, we swapped them, creating SUMO1-E93Q and SUMO2-Q89E. Remarkably, the inclusion of these substitutions switched the conjugation bias of SAE1:SAE2-K164Q from SUMO2 to SUMO1 (Fig. 2F). These data are consistent with the idea that residue 164 of SAE2 contributes to discriminative interactions with SUMO protein C-terminal regions, influencing SUMO protein variant conjugation.

## SAE1:SAE2-K164 supports mitotic fidelity

To investigate the cellular consequences of SAE2 K164 modification, we generated a dual-expression system for siRNA-resistant SAE1:-SAE2. Constructs included wild-type (WT) E1, an acetyl-analogue variant (SAE1:SAE2-K164Q), a thiolation-inactive mutant (SAE1:-SAE2-C173G) (Desterro et al, 1999; Johnson et al, 1997; Okuma et al, 1999) and a structurally conservative, non-modifiable variant (SAE1:SAE2-K164R) (Fig. EV4A). The latter was designed to maintain charge similarity and test the degree to which lysine 164 modification, which is lost in both K164Q and K164R mutations, is needed.

We found that none of these mutations altered the sub-cellular localisation of SAE2 (Fig. EV4B). To assess their impact on SUMO-dependent processes, we first examined cell survival following heat shock, a stress that requires widespread SUMOylation (Golebiowski et al, 2009). SAE1:SAE2 depletion sensitised cells to 43 °C exposure, a phenotype that was rescued by expression of WT, K164R, or K164Q SAE1:SAE2, but not the catalytic mutant C173G (Fig. EV4C), suggesting that K164 modification is dispensable for the SUMO-mediated heat shock response. It may be relevant that the heat shock response involves the conjugation of SUMO2/3 over SUMO1 (Golebiowski et al, 2009; Pinto et al, 2012).

The cellular response to DNA double-strand breaks requires many SUMOylation events (Garvin and Morris, 2017). As expected, cells exposed to irradiation showed delayed resolution of the DNA damage marker γH2AX on depletion of UBC9 or the SUMO E3 ligase PIAS1 (Fig. EV4D). Intriguingly, however, we found no impact of SAE1:SAE2 depletion or SAE2-complementation, even with SAE1:SAE2-C173G, on γH2AX kinetics (Fig. EV4D). Consistent with these observations, we observed only a slight impact of SAE1:SAE2 depletion on the repair of enzymatically generated DNA double-strand breaks (Fig. EV4E). These results are surprising in light of the requirement for SUMOylation in the DNA damage response. Nevertheless, they

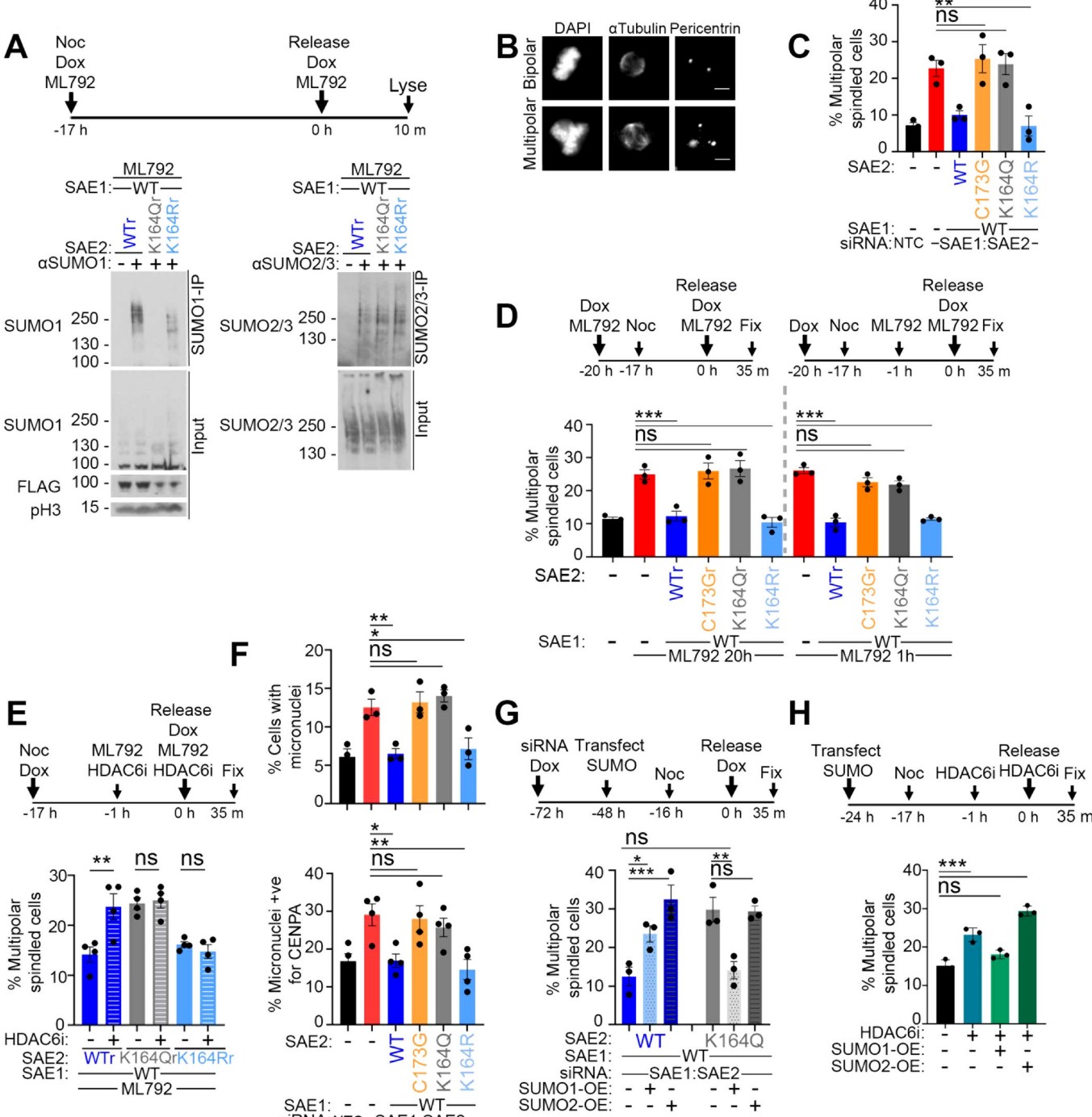

are consistent with the report that SUMO E1 inhibition has no impact on the cellular response to agents causing DNA damage; cisplatin or hydroxyurea (He et al, 2017).

We next considered the potential influence on mitosis, as the HDAC6:SAE2 interaction and acK164-SAE2 deacetylation coincided with early mitosis (Fig. 1A,B). To examine the impact of SAE1:SAE2 on mitosis more precisely, we generated a second complementation system to allow acute E1 inhibition. In this system, U2OS cells express a SAE2 mutant, SAE2-S95N-M97T, that

we name "FLAG-SAE2r", which provides resistance to the SUMO E1 inhibitor, ML792 (He et al, 2017). Into FLAG-SAE2r, we added SAE2-C173G, SAE2-K164Q and SAE2-K164R mutations. We assessed the impact of FLAG-SAE2r-K164 mutations on mitotic SUMOylation by synchronising cells in prometaphase with nocodazole with the concurrent addition of ML792, examining cells 10 min after release from nocodazole treatment. From complemented, mitotic cells, we immunoprecipitated SUMO proteins. High-molecular-weight SUMO1 was almost undetectable

◄  **Figure 3.  K164-SAE2 directs mitotic fidelity.**

(**A**) Immunoprecipitation of endogenous mitotic SUMO conjugates in U2OS cells treated with 1 μM ML792 and expressing FLAG-SAE2 constructs resistant to the inhibitor. ML792 resistance is denoted by (r). The presence of FLAG-SAE2r is represented by (WTr), FLAG-SAE2r-K164Q by K164Qr, and FLAG-SAE2r-K164R by K164Rr. Diagram, top, illustrates the timing of inhibitor and induction agent addition. Antibodies for SUMO proteins were EP298 (SUMO1) and 8A2 (SUMO2/3). Performed once. (**B**) Representative images of immunofluorescent analysis of mitotic spindle formation. Five micrometer scale bar is shown as a white line. (**C**) The percent of laterally presented metaphase and anaphase U2OS cells exhibiting multipolar spindles in cells complemented with SAE2 variants in RNAi-treated cells, as shown. Error bars SEM; significance calculated using one-way ANOVA, *$P \le 0.05$, **$P \le 0.01$, ns = not significant >0.05. siSAE:1:SAE2 vs WT $P = 0.0142$, siSAE:1:SAE2 vs CG $P = 0.9034$, siSAE:1:SAE2 vs KQ $P = 0.9964$, siSAE:1:SAE2 vs KR = 0.0031. Data from three independent biological repeats. $N > 50$ cells per condition analysed from a minimum of four fields of view per biological repeat. (**D**) The percent of laterally presented metaphase and anaphase U2OS cells exhibiting multipolar spindles. Cells exposed to different durations of 1 μM ML792 prior to release from 0.332 μM nocodazole: long treatment totalling 20 h (added 4 h before nocodazole) is displayed to the left-hand side, short treatment totalling 1 h (added during the last hour of nocodazole), presented to the right-hand side. 1 μM ML792 was re-added to cells after release from nocodazole. Error bars SEM; significance calculated using one-way ANOVA, ***$P \le 0.001$, ns = not significant >0.05. ML792 20 h vs WTr $P = 0.0002$, ML792 20 h vs CGr $P > 0.9999$, ML792 20 h vs KQr $P = 0.9982$, ML792 20 h vs KRr $P < 0.0001$. ML792 1 h vs WTr $P < 0.0001$, ML792 1 h vs CGr $P = 0.8286$, ML792 1 h vs KQr $P = 0.6389$, ML792 1 h vs KRr $P < 0.0001$. Data from 3 independent biological repeats. $N > 50$ cells per condition from a minimum of four fields of view per biological repeat. (**E**) The percent of laterally presented metaphase and anaphase U2OS cells exhibiting multipolar spindles in cells treated with 1 μM ML792 and complemented with WTr, FLAG-SAE2r-K164Q, and FLAG-SAE2r-K164R with and without the addition of HDAC6 inhibitor. Error bars SEM; significance calculated using one-way ANOVA, **$P \le 0.01$, ns = not significant >0.05. WTr + vehicle vs WTr + HDAC6i $P = 0.0050$, KQr + vehicle vs KQr + HDAC6i $P = 0.9998$, KRr + vehicle vs KRr + HDAC6i $P = 0.9883$. Data from four independent biological repeats. $N > 50$ cells per condition from a minimum of four fields of view per biological repeat. (**F**) Analysis of micronuclei in asynchronous siRNA-resistant SAE2 variant U2OS cells. The percentage of total cells with one or more micronuclei is plotted (Top). Error bars SEM; significance calculated using one-way ANOVA, *$P \le 0.05$, **$\le 0.01$, ns = not significant >0.05. Data from three independent biological repeats. Cells from at least 4 fields of view were analysed. $N > 400$ total cells per condition per biological repeat. siSAE:1:SAE2 vs WT $P = 0.0083$, siSAE:1:SAE2 vs CG $P = 0.9893$, siSAE:1:SAE2 vs KQ $P = 0.7940$, siSAE:1:SAE2 vs KR $P = 0.0176$. The percentage of micronuclei positive for CENPA in asynchronous siRNA-resistant SAE2 variant cells. (Bottom): Error bars SEM; significance calculated using one-way ANOVA. *$P \le 0.05$, **$\le 0.01$, ns = not significant >0.05. Data from four independent biological repeats. $N > 50$ micronuclei per condition per biological repeat. siSAE:1:SAE2 vs WT $P = 0.0163$, siSAE:1:SAE2 vs CG $P = 0.9986$, siSAE:1:SAE2 vs KQ $P = 0.8261$, siSAE:1:SAE2 vs KR $P = 0.0041$. (**G**) The percent of metaphase U2OS cells with multipolar spindles in cells expressing siRNA-resistant SAE2 variants after release from 0.332 μM nocodazole, with or without SUMO1 or SUMO2 overexpression. Timeline of the experiment depicted above. Error bars SEM; significance calculated using one-way ANOVA. *$P \le 0.05$, **$P \le 0.01$, ns = not significant >0.05. Data from three independent biological repeats. $N > 50$ cells per condition from at least four fields of view per biological repeat. WT vs WT + SUMO1-OE $P = 0.0217$, WT vs WT + SUMO2-OE $P = 0.0007$, WT vs KQ + SUMO1-OE $P = 0.6593$, KQ vs KQ + SUMO1-OE $P = 0.0045$, KQ vs KQ + SUMO2-OE $P = 0.9117$. (**H**) Mean percentage of metaphase U2OS cells with multipolar spindles in cells treated with HDAC6 inhibitor, with or without 24 h SUMO1 or SUMO2 overexpression. Timings of the experiment are displayed above. In total, 2.5 μM HDAC6 inhibitor was added 1 hr prior to release from 0.332 μM nocodazole and replaced onto cells for the duration of mitotic release. Error bars SEM; significance calculated using one-way ANOVA, *$P \le 0.05$, **$P \le 0.01$, ***$P \le 0.001$, ns = not significant >0.05. Data from three independent biological repeats. $N > 50$ cells per condition from at least four fields of view per biological repeat. Vehicle vs HDAC6i $P = 0.0004$, Vehicle vs HDAC6i + SUMO1-OE $P = 0.0900$, Vehicle vs HDAC6i + SUMO2-OE $P < 0.0001$.

in FLAG-SAE2r-K164Q complemented cells, but prevalent in FLAG-SAE2r-K164R complemented cells when FLAG-SAE2r-K164R and FLAG-SAE2r-K164Q were expressed at an equivalent, but lower than WT FLAG-SAE2r level (Fig. 3A). Using an antibody better able to detect free SUMO1 (Garvin et al, 2022), we noted that FLAG-SAE2r-K164Q expressing cells had slightly more free SUMO1 (Fig. EV4F). The level of SUMO2/3 precipitates was comparable from cells complemented with either FLAG-SAE2r-K164Q, FLAG-SAE2r-K164R, or FLAG-SAE2r, despite the lower expression levels of the K164-mutant proteins (Fig. 3A). These data suggest that mitotic SUMO1 conjugate generation is sensitive to E1 levels and to SAE2-K164Q.

SUMO conjugation promotes correct spindle assembly and chromosome segregation (Abrieu and Liakopoulos, 2019; Mukhopadhyay and Dasso, 2017). To assess whether SAE2-K164 impacts mitotic spindle assembly, we depleted endogenous SAE1:SAE2 and expressed siRNA-resistant SAE1:SAE2 variants before synchronising with nocodazole, washing out, and immunostaining for components of the mitotic spindle machinery, α-tubulin and pericentrin, inspecting cells in metaphase and anaphase. Depletion of SAE1:SAE2 increased the proportion of cells with multipolar spindles, which were suppressed by the expression of SAE1:SAE2 or SAE1:SAE2-K164R (Fig. 3B,C). However, strikingly, neither SAE1:SAE2-C173G nor SAE1:SAE2-K164Q expression suppressed multipolar spindle formation (Fig. 3C). Thus, SAE2-K164Q impairs bipolar spindle formation, whereas SAE2-K164R supports normal spindle polarity.

To begin addressing how spindle defects might arise, we inquired about the timing of E1 activity. Multipolar spindle

formation can be driven by centrosome amplification in S-phase or G2, and also by aberrant initiation of spindle assembly occurring on or after nuclear envelope breakdown (Maiato and Logarinho, 2014). We tested FLAG-SAE2r expressing cells and added the E1 inhibitor in the last hour of nocodazole treatment, before wash-out, as well as before nocodazole exposure (Fig. 3D). Under both conditions, FLAG-SAE2r-WT and FLAG-SAE2r-K164R, but not FLAG-SAE2r-C173G or FLAG-SAE2r-K164Q, suppressed the formation of multipolar spindles (Fig. 3D). These data discount defects in S-phase or early G2 as the drivers of multipolar spindle generation. Instead, they suggest that the absence of SUMOylation and the impact of SAE2-K164Q disrupt a process required directly before or during the re-polymerisation of the microtubule network.

Next, we tested the role of HDAC6 deacetylation activity in the formation of multipolar spindles in the context of FLAG-SAE2r variants. Cells containing FLAG-SAE2r variants were synchronised with nocodazole for 16 h and treated for the last 1 h with ML792 with or without HDAC6 inhibitor, ACY-738, before releasing into mitosis (Fig. 3E). HDAC6 inhibition resulted in a significant increase in multipolar spindles in the FLAG-SAE2r-WT cells, consistent with previous reports of HDAC6 inhibition (Huang et al, 2017). Notably, HDAC6 inhibition had no significant effect on multipolar spindle levels in cells expressing either K164 mutant (Fig. 3E). Multipolar spindle frequency remained low in FLAG-SAE2r-K164R–expressing cells and remained high in FLAG-SAE2r-K164Q–expressing cells, even after HDAC6 inhibitor treatment (Fig. 3E). These results indicate that SAE2-K164 mutations confer insensitivity to HDAC6 inhibition, supporting

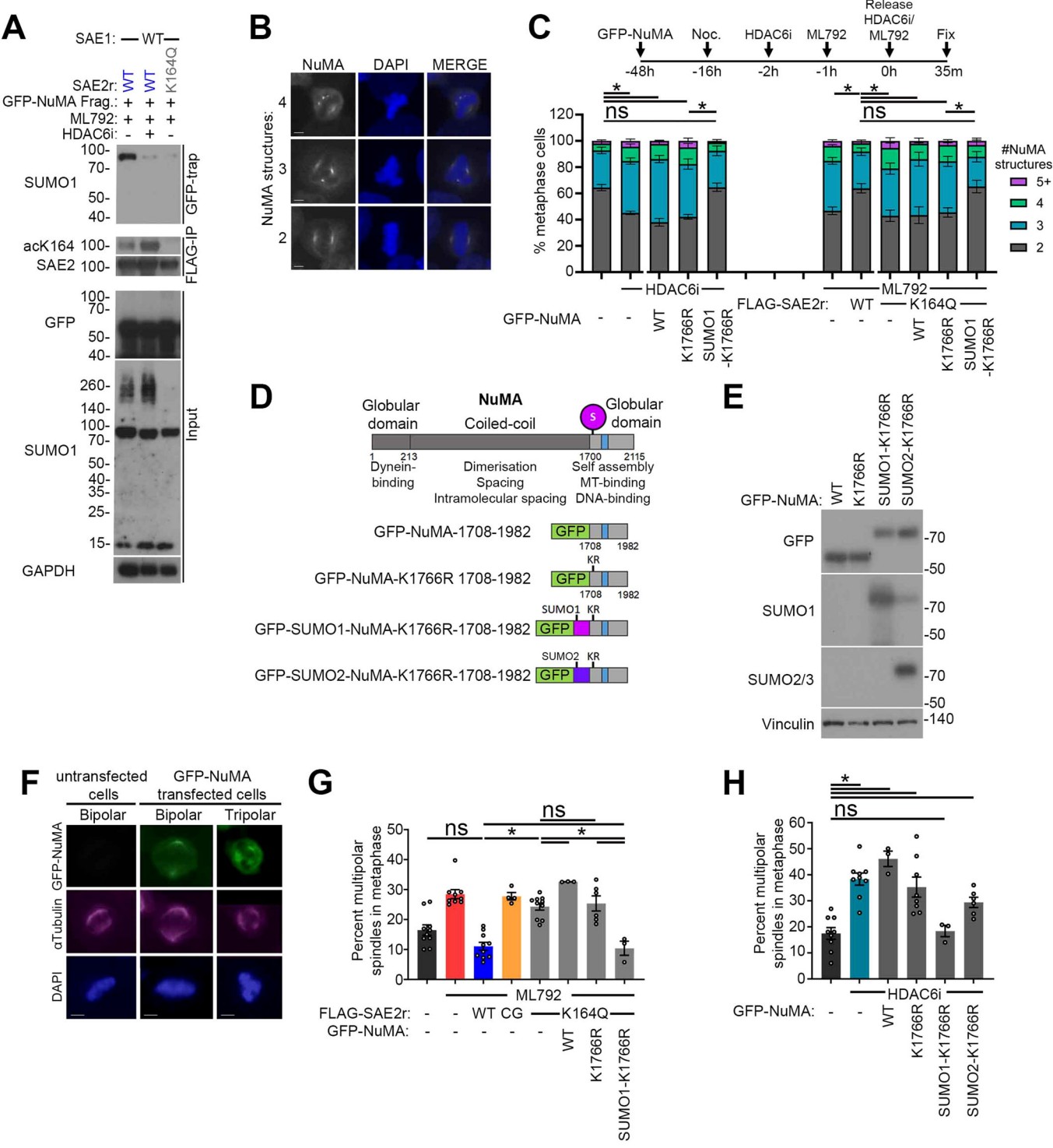

the notion that acetylation and deacetylation of SAE2-K164 regulates spindle assembly fidelity.

Cells with more than two spindle poles may segregate chromosomes poorly, and misaligned, lagging, or bridge chromosomes can become isolated as encapsulated DNA fragments, known as micronuclei (Cosper et al, 2022). We examined SAE1:SAE2 siRNA-complemented cells for the presence of micronuclei. We found that

SAE1:SAE2 and SAE1:SAE2-K164R complementation, but not SAE1:SAE2-C173G or SAE1:SAE2-K164Q, suppressed increased numbers of micronuclei (Fig. 3F, top panel), consistent with the notion that inappropriate spindle formation results in micronuclei in these cells. To gain further insight into the contents of the micronuclei, we stained for the centromere component CENPA and the DNA-damage marker γH2AX. We observed no enrichment for γH2AX-

**Figure 4. Mitotic defects incurred by SAE2-K164Q or HDAC6 inhibition are corrected by GFP-SUMO1-NuMA fragment expression.**

(A) Western blot analysis of U2OS expressing FLAG-SAE2r-WT transfected with a GFP-NuMA fragment treated with 0.332 μM nocodazole and 5 μM ML792 ± 5 μM ACY-738 (HDAC6i) for 16 h, alongside FLAG-SAE2r-K164Q expressing cells (5 μM ML792). Mitotic cells were harvested by mitotic shake-off, lysed, and subjected to GFP-trap and western blots probed for SUMO1. The lysates were also precipitated by anti-FLAG beads and probed with antibodies for acK164-SAE2 and SAE2, while inputs were probed for GFP, SUMO1, and GAPDH. Replicated three times in the laboratory. (B) Example images of 2, 3, and 4 NuMA structures in metaphase U2OS cells immunostained for NuMA and with Hoechst. White bar indicates 5 micrometers. (C) Average percentage of the metaphase cell population with 2 (grey), 3 (blue), 4 (green), and 5+ (purple) NuMA structures in untransfected U2OS cells treated with 2.5 μM HDAC6i (left-hand side of the graph) or cells expressing FLAG-SAE2r-K164Q and treated with 1 μM ML792 (right-hand side of the graph). Cells were also transfected with c-terminal fragments of GFP-NuMA, GFP-NuMA-K1766R, and GFP-SUMO1-NuMA-K1766R. Means are plotted with error bars as SEM for $N = 4$ independent repeats ( > 50 cells counted per condition) with one-way ANOVA used to assess the statistical significance for the percentage of metaphase cells with 2 NuMA structures, where $*P < 0.05$ and ns $= P > 0.05$. Statistical values are - vs HDAC6i $P < 0.0001$, - vs HDAC6i + GFP-NuMA $P < 0.0001$, - vs HDAC6i + GFP-NuMA-K1766R $P < 0.0001$, - vs HDAC6i + GFP-SUMO1-NuMA-K1766R $P > 0.9999$, HDAC6i + GFP-NuMA-K1766R vs HDAC6i + GFP-SUMO1-NuMA-K1766R $P = 0.0009$, ML792 vs ML792 + SAE2-WT $P < 0.0001$, ML792 + SAE2-WT vs ML792 + SAE2-K164Q $P < 0.0001$, ML792 + SAE2-WT vs ML792 + SAE2-K164Q + GFP-NuMA $P = 0.0002$, ML792 + SAE2-WT vs ML792 + SAE2-K164Q + GFP-NuMA-K1766R $P < 0.0001$, ML792 + SAE2-WT vs ML792 + SAE2-K164Q + GFP-SUMO1-NuMA-K1766R $P > 0.9999$, ML792 + SAE2-K164Q + GFP-NuMA-K1766R vs ML792 + SAE2-K164Q + GFP-SUMO1-NuMA-K1766R $P = 0.0008$. (D) Diagram of NuMA monomer indicating the dominant SUMO1ylation site (purple sphere) at K1766 in the C-terminal 'self-assembly' domain (Seo et al, 2014). Shown below are the GFP-NuMA C-terminal constructs, GFP-NuMA-1708-1982, GFP-NuMA-K1766R-1708-1982, and GFP-SUMO1-NuMA-K1766R-1708-1982 and GFP-SUMO2-NuMA-K1766R-1708-1982 linear fusions. (E) Western blot for the GFP-tag showing U2OS transfected with GFP-NuMA constructs. Performed once. (F) Representative metaphase cell images of U2OS cells expressing GFP-NuMA, immunostained for α-tubulin, showing bipolar and multipolar (3) spindles. White bar indicates 5 μm. (G) Mean percentage of the metaphase cell population with multipolar spindles in U2OS expressing FLAG-SAE2r-K164Q treated with 1 μM ML792 with WT or K1766R C-terminal fragment NuMA variants or SUMO1 fused C-terminal K1766R- NuMA fragment and stained for α-tubulin. $N = 9$ biological repeats, bars $=$ SEM, and statistical significance was calculated using one-way ANOVA where $*P < 0.05$, and ns $= P > 0.05$. Statistical values for - vs SAE2-WT $P = 0.1402$, ML792 vs ML792 + SAE2-WT $P < 0.0001$, ML792 + SAE2-WT vs ML792 + SAE2-K164Q + GFP-SUMO1-NuMA-K1766R $P > 0.9999$, ML792 + SAE2-K164Q vs ML792 + SAE2-K164Q + GFP-NuMA $P = 0.0290$, ML792 + SAE2-K164Q vs ML792 + SAE2-K164Q + GFP-NuMA-K1766R $P = 0.9482$, ML792 + SAE2-K164Q vs ML792 + SAE2-K164Q + GFP-SUMO1-NuMA-K1766R $P = 0.0004$, ML792 + SAE2-K164Q + GFP-NuMA-K1766R vs ML792 + SAE2-K164Q + GFP-SUMO1-NuMA-K1766R $P = 0.0042$. (H) Average percentage of the metaphase cell population with multipolar spindles as assessed by α-tubulin structures in U2OS cells treated with 2.5 μM HDAC6 inhibitor, ACY-738, transfected with GFP-C-terminal fragments, GFP-SUMO1-NuMA-K1766R or GFP-SUMO2-NuMA-K1766R fusion constructs. $N = 7$ independent biological experiments, bars $=$ SEM, and statistical significance calculated using one-way ANOVA where $*P < 0.05$ and ns $= P > 0.05$. Statistical values are – vs HDAC6i $P < 0.0001$, – vs HDAC6i + GFP-NuMA $P = 0.0306$, – vs HDAC6i + GFP-NuMA-K1766R $P = 0.0010$, – vs HDAC6i + GFP-SUMO1-NuMA-K1766R $P = 0.9329$, – vs HDAC6i + GFP-SUMO2-NuMA-K1766R $P = 0.0075$.

containing micronuclei in SAE1:SAE2 depleted or complemented cells (Fig. EV4G), suggesting no increased chromosome fragments in the micronuclei observed. However, approximately 1/3rd of micronuclei in SAE1:SAE2 depleted or SAE1:SAE2-C173G or SAE1:SAE2-K164Q complemented cells showed staining for CENPA (Fig. 3F, bottom panel), suggesting a proportion of the micronuclei contain centric chromosomes.

Given that SAE2 K164 promotes SUMO1 conjugation under conditions of SUMO1:SUMO2 competition (Fig. 2D), and that SUMO1-conjugates in mitotic SAE2-K164Q-complemented cells are suppressed (Fig. 3A), we next asked whether multipolar spindle defects could be induced or repressed by altering the relative expression of SUMO variants. Remarkably, SUMO1, but not SUMO2, overexpression suppressed the formation of multipolar spindles in SAE1:SAE2-K164Q complemented cells (Fig. 3G), consistent with the notion that SUMO1ylation driven by SAE2-K164 is critical to spindle regulation. Intriguingly, both SUMO1 and, in particular, SUMO2 overexpression also increased the number of multipolar spindles in cells complemented with WT SAE1:SAE2, suggesting that disrupting SUMO variant balance disturbs spindle assemblies. Further, consistent with the idea that HDAC6 drives SAE2 deacetylation to support mitotic SUMO1ylation, SUMO1 overexpression also suppressed multipolar spindle formation in cells treated with HDAC6 inhibitor (Fig. 3H).

## SUMO1-NuMA fusion suppresses spindle defects in HDCA6-inhibitor-treated and SAE2-K164Q-complemented cells

Our findings of a SAE2-K164:SUMO1 dependency in mitotic spindle organisation led us to consider whether known

SUMO1ylated substrates explain the defect in HDAC6-suppressed or SAE1:SAE2-K164Q-complemented cells. SUMO1 conjugates RanBP2/RanGAP1-SUMO1/UBC9 complex and nuclear mitotic apparatus (NuMA) have been previously associated with the promotion of bipolar mitotic spindles (Flotho and Werner, 2012; Seo et al, 2014). The remainder of the known SUMO1 conjugates in mitosis, BubR1, Aurora-A, and PLK1, promote mitotic timing or microtubule polymerisation (Perez de Castro et al, 2011; Wen et al, 2017; Yang et al, 2012). RanGAP1-SUMO1 remains stable over several cell cycles upon treatment with ML792 (He et al, 2017) or siUBC9 (Hayashi et al, 2002). Thus, we reasoned that downregulated RanGAP1-SUMO1 is unlikely to be responsible for the mitotic defects observed following acute E1 suppression.

Immunoblotting for NuMA in mitotic cells revealed bands at the expected molecular weight (~238 kDa) and a higher band at ~250 kDa. The latter was abolished by treatment with the E1 inhibitor ML792 (Fig. EV4H), consistent with the previously reported mitotic SUMO1ylation of NuMA (Seo et al, 2014). To test whether SUMOylation of NuMA can be affected by SAE-regulation, we transfected cells expressing FLAG-SAE2r variants with a construct expressing GFP-NuMA[1708-1982] and treated with ML792, or ML792 and HDAC6i (Fig. 4A). Amino acids 1708-1982 of NuMA bear the major mitotic SUMOylation site, mapped to lysine-1766 (Seo et al, 2014). Mitotic cell lysates were incubated with beads conjugated to an anti-GFP nanobody, and the "GFP-trapped" material was investigated by immunoblot. GFP-enriched proteins from ML792-treated cells expressing FLAG-SAE2r-WT produced a SUMO1 band at ~80 kDa, indicative of SUMO1 conjugation onto GFP-NuMA[1708-1982] (Fig. 4A). Co-treatment with HDAC6 inhibitor upregulated acK164-SAE2 of FLAG-SAE2-WT and severely reduced the level of SUMO1 co-purified by GFP-

NuMA$^{1708-1982}$. Complementation of ML792 with FLAG-SAE2r-K164Q similarly suppressed the ~80 kDa SUMO1 co-purified by GFP-NuMA$^{1708-1982}$ (Fig. 4A). These data indicate mitotic NuMA-SUMO1ylation depends upon HDAC6 deacetylase activity and is suppressed by SAE2-K164Q, consistent with a requirement for acK164-SAE2 deacetylation for modification.

NuMA aids the clustering of the microtubule fibre minus ends at the spindle poles around the centromere in early mitosis, and NuMA dysfunction causes spindle pole-focusing defects, lagging chromosomes in anaphase and micronuclei formation (Chinen et al, 2020; Kiyomitsu and Boerner, 2021; van Toorn et al, 2023). The SUMOylation-deficient NuMA mutant, K1766R, is defective in recruitment to spindle poles and in microtubule bundling, as a result, multipolar spindles are induced during mitosis (Seo et al, 2014).

To determine whether SAE2 regulation is critical to SUMO-related activities of NuMA, we assessed NuMA assemblies (confirming the specificity of the NuMA antibody in Fig. EV4I). We inspected NuMA in HDAC6-inhibitor and ML792-treated and complemented metaphase cells, counting observed structures (Fig. 4B). >2 indicates disordered spindle organisation (Chinen et al, 2020; van Toorn et al, 2023). HDAC6-inhibitor treatment reduced the proportion of cells bearing just 2 NuMA structures and increased the incidence of >2 assemblies (Fig. 4C left-hand side). This observation was also made in cells treated with ML792 (Fig. 4C, right-hand side). Importantly, in ML792-treated cells, complementation with FLAG-SAE2r-WT, but not FLAG-SAE2r-K164Q, was able to restore the percentage of cells with 2 NuMA structures to untreated levels and reduced the number of cells with >2 NuMA structures (Fig. 4C, right-hand side). Then, to investigate whether the direct regulation of NuMA SUMOylation is related to NuMA structure and spindle defects in HDAC6i-treated cells and cells complemented with FLAG-SAE2r-K164Q, we expressed a series of NuMA constructs (Fig. 4D,E). Since many of the functional roles of the protein are encoded in the globular C-terminus, and a C-terminal fragment can perform the mitotic roles of full-length NuMA (Seo et al, 2014), we expressed the GFP-tagged NuMA$^{1708-1982}$ with and without the K1766R mutation. Additionally, we tested this mutant GFP-NuMA fragment carrying SUMO1 fused between the GFP and NuMA fragment, to mimic SUMO1ylated-K1766-NuMA (Fig. 4C–E), as previously described (Seo et al, 2014). Expression of the WT or K1766R mutant C-terminal NuMA fragment had little impact on NuMA structure numbers in HDAC6 inhibitor-treated cells or in ML792-treated cells complemented with FLAG-SAE2r-K164Q, where cells with >2 structures remained high (Fig. 4C). Remarkably however, expression of the construct bearing the SUMO1 fusion, GFP-SUMO1-NuMA-K1766R, suppressed NuMA structures in both contexts, resulting in the majority of HDAC6-inhibitor and the majority of FLAG-SAE2r-K164Q complemented cells exhibiting just 2 NuMA structures (Fig. 4C). Thus, the expression of a SUMO1-NuMA fusion can prevent both the harmful impact of HDAC6 inhibition and complementation with SAE2-K164Q, to support bipolar NuMA structures.

We then tested GFP-NuMA constructs for their ability to suppress multipolar spindles, assessed by α-tubulin (Fig. 4F), in HDAC6 inhibitor-treated or FLAG-SAE2r-K164Q complemented cells. As for NuMA structures, we found bipolar spindles were restored and multipolar spindles reduced in cells expressing the GFP-SUMO1-NuMA-K1766R fragment (Fig. 4G,H). Thus, the SUMO1-NuMA fusion can also improve bipolar spindle formation in conditions of HDAC6 inhibition or FLAG-

SAE2r-K164Q complementation. Finally, to test whether there is a particular requirement for SUMO1, we introduced GFP-SUMO2-NuMA-K1766R (Fig. 4D–E,H). GFP-SUMO2-NuMA-K1766R failed to correct the HDAC6 inhibitor-induced multipolar spindles (Fig. 4H), indicating that SUMO1-modification of NuMA, but not SUMO2-, promotes bipolar spindle arrangement.

## Discussion

The SUMO E1 enzyme differs from most other ubiquitin-like modifier-activating enzymes in that it activates related but different modifiers. Here, we describe a mechanism by which the SUMO E1 can direct SUMO protein conjugation bias. In our in vitro assays, an acetylation-mimic, SAE1:SAE2-K164Q, displays accumulation of SUMO1-adenyl, a reduced ability to form the SAE2~SUMO1 thioester, and reduced RanGAP1-SUMO1ylation and increased RanGAP1-SUMO2ylation in reactions containing SAE1:SAE2-K164Q and SUMO1 and SUMO2. The SAE2-K164Q-bearing E1 enzyme discriminates between SUMO variants through the SUMO C-terminal tail residues, SUMO1-E93 and SUMO2-Q89. Our findings are similar to the discrimination that the NEDD8 E1 (APPBP1–UBA3) enzyme employs to maintain the specificity of NEDD8 over the 55% identical ubiquitin. APPBP1–UBA3 selectively binds NEDD8-A72, but not ubiquitin-R72 (Walden et al, 2003). Walden et al (2003) speculated that other E1 enzymes may employ similar modifier discrimination, and indeed NEDD8-A72 and Ubiquitin-R72 residues align with SUMO1-E93 and SUMO2-Q89 (Fig. EV5A).

In structural assessments of the SUMO E1 with SUMO, the E1 assumes an adenylation catalysing 'open' conformation and a thioester bond catalysing 'closed' conformation (Lois and Lima, 2005; Olsen et al, 2010). The 'open' conformation suggests hydrogen bonding between SAE2-R119-Y159 and SUMO1-E93 (Lois and Lima, 2005), with the 'closed' conformation exhibiting a closer association and potential hydrogen bond between SAE2-K164 and SUMO1-E93/SUMO2-Q89 (Hann et al, 2019; Olsen et al, 2010) (Fig. EV5B). Additionally, an electrostatic interaction between SAE2-K164 and SUMO1-E93 may contribute. A model where SAE2-K164 acetylation reduces hydrogen-bonding and electrostatic interaction with SUMO1-E93 in the 'closed' SUMO E1 conformation is consistent with our data, suggesting SAE1:-SAE2-K164Q adenylates SUMO1 and squanders SUMO1-AMP before SAE2~SUMO1 thioester formation. SAE2-acK164 may acquire hydrogen bond acceptor capacity to which SUMO2-Q89 donates a hydrogen bond, while SUMO1-E93 cannot, and any electrostatic interaction between K164-SAE2 and E93-SUMO1 is lost on K164 acetylation (Fig. EV5B). The observation that SAE2-K164Q enhances SUMO2 preference even without ATP implies that the pre-adenylation conformation of SAE1:SAE2 also contributes to SUMO variant selectivity. We speculate that both the 'open' and 'closed' conformations of the SUMO E1 enzyme coexist in solution.

Our data do not support a positive role for K164 modification, whether acetylated or carrying another modification, as the K164R-SAE2 mutant exhibits a minimal phenotype in our hands. Nevertheless, K164 is highly conserved (e.g., in *S. cerevisiae*, *D. melanogaster*, *D. rerio*, and *H. sapiens*), so we do not discount the possibility that acetylation of K164 is employed to favour SUMO2/3ylation in other pathways we have not tested. We also do not rule

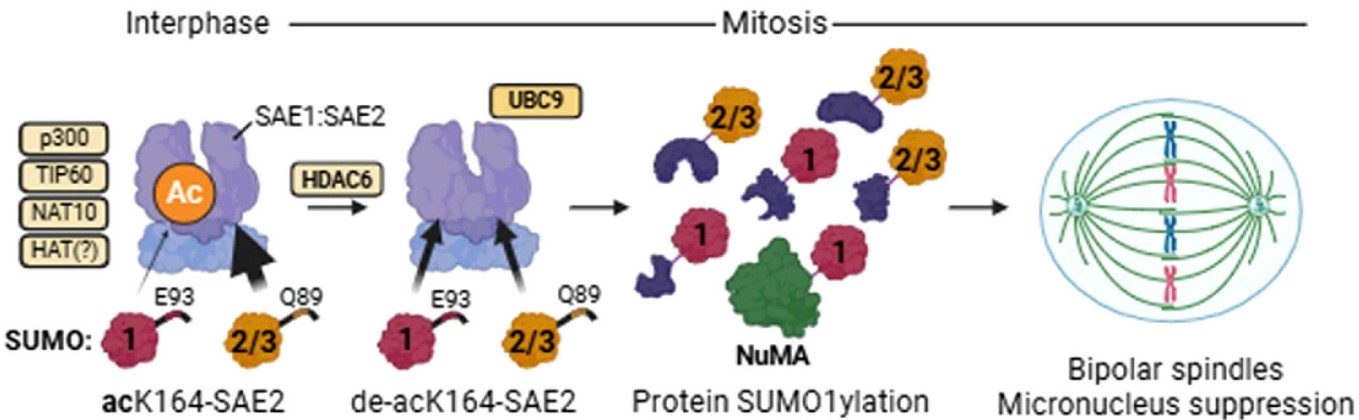

**Figure 5. SAE1:SAE2-acK164 is downregulated in mitosis to promote mitotic fidelity.**

SAE1:SAE2-acK164 is present in mammalian cells due to histone-acetyltransferase activities including p300, TIP60, and NAT10. SAE1:SAE2-acK164 biases the activation of SUMO2 over SUMO1. During mitosis, SAE1:SAE2-acK164 is downregulated in an HDAC6-dependent manner to improve activation of SUMO1 and enable protein SUMO1ylation including NuMA-SUMO1 formation, to promote bipolar mitotic spindles and genomic stability. Created with Biorender.

out other regulatory modifications of the site, since modification with ubiquitin/SUMO would be expected to have an inhibitory impact on activity.

We show that the SUMO E1 enzyme is deacetylated at K164-SAE2 after ionising radiation and in cells synchronised in early mitosis. We find no role for the E1 in double-strand DNA break repair or in modulating markers of DNA damage after irradiation, and instead find a clear association with mitotic fidelity. The deacetylation of acK164-SAE2 can be suppressed by HDAC6 inhibition, and our results strongly align HDAC6 activity with SAE2-K164, as mutation of K164 can overcome the impact of HDAC6 inhibition. Intriguingly, HDAC6 is one of the few histone deacetylases found primarily in the cytoplasm, where it catalyses the removal of acetyl groups from substrates, including α-tubulin and HSP90 (Asthana et al, 2013; Boyault et al, 2007). As the SUMO E1 enzyme localisation is predominantly nuclear (Azuma et al, 2001), we speculate that the interaction between acK164-SAE2 and HDAC6 is increased by nuclear envelope breakdown at the end of prophase in mitosis. Remarkably, the defects of increased multipolar spindles observed upon HDAC6 inhibition can be rescued by the expression of the SUMO1-NuMA fusion, but not by SUMO2-NuMA, suggesting that much of the role of HDAC6 in supporting a bipolar mitotic spindle relates to SUMO1ylation of NuMA.

SUMO protein availability and the relative kinetic properties of the SUMO-activating enzyme are likely to drive SUMO variant protein bias. SAE1:SAE2 exhibits a higher affinity (lower Km) for SUMO1 than for SUMO2/3, but it also has a lower kcat, resulting in an approximately equal kcat/Km for both SUMO variants. Consequently, under equal SUMO1 and SUMO2/3 concentrations, SAE1:SAE2 facilitates nearly equivalent conjugation of SUMO1 and SUMO2/3 (Lois and Lima, 2005; Wiryawan et al, 2015, and this study). However, at low SUMO concentrations, the lower Km for SUMO1 would be expected to drive SUMO1 conjugation, despite its lower kcat. In contrast, at high concentrations, or altered relative concentrations, SUMO2/3 > SUMO1, the higher Km for SUMO2/3 initially makes it less competitive, but the higher Kcat drives rapid turnover. During mitosis, total SUMO protein concentrations are lower compared to those in asynchronous cells, coinciding with

nuclear envelope breakdown (Zhang et al, 2008). If free SUMO1 and SUMO2/3 levels were equal at the lower end of the kcat/Km curve, the reduced SUMO availability would be expected to favour SUMO1 activation. However, SUMO2/3 constitutes the majority (>90%) of the total SUMO pool (Saitoh and Hinchey, 2000) and SUMO2/3 conjugates continue to dominate over SUMO1-modified proteins during mitosis (Zhang et al, 2008). Our findings suggest that retention of acetylated K164-SAE2 in mitosis (evidenced through HDAC6 inhibition or K164Q-SAE2 complementation) further reduces the likelihood of SUMO1 activation. Therefore, we propose that E1 deacetylation helps maintain the limited mitotic SUMO1ylation. A limitation of our study is that the amount of acetylated SAE2 is unclear. We have detected a low signal using the acK164 antibody on endogenous SAE2, and we anticipate that acK164 levels are low.

SAE2-K164 suppression of abnormal multipolar spindles and CENPA-positive micronuclei aligns with previous findings that SUMO E1 depletion or inhibition drives mitotic defects (Eifler et al, 2018; He et al, 2017). Our findings also provide a mechanistic explanation for the previous observation that NuMA-dependent multipolar spindles are induced by HDAC6 inhibition (Huang et al, 2017). We propose a model in which SAE2 K164 deacetylation by HDAC6 during mitosis promotes SUMO1 conjugation, specifically enhancing SUMO1ylation of NuMA to facilitate its clustering and support of bipolar spindle formation (Fig. 5).

NuMA has both mitotic and interphase roles. In interphase, a form of NuMA missing one of its C-terminal microtubule-binding regions can nevertheless contribute to single-stranded DNA break repair (Ray et al, 2022). In contrast, a NuMA C-terminal fragment is sufficient to establish bipolar spindles in metaphase cells (Seo et al, 2014). The defect of extra NuMA structures and multipolar spindles in conditions of HDAC6 inhibition or SAE2-K164Q complementation is suppressed by the linear GFP-SUMO1-NuMA, but not by GFP-SUMO2-NuMA. While we do not discount the requirement for other acutely SUMO1ylated mitotic substrates in SAE2-K164Q complemented cells or HDAC6-inhibited cells, our data suggest SUMO1-NuMA can accomplish much of the role(s) that any other SUMO1-substrate(s) might perform in supporting bipolar mitotic spindle assembly.

The major NuMA SUMO1ylation site at K1766, within P-K-V-E, (a SUMO conjugation consensus site, ψ-K-x-E) that overlaps with NuMA's clustering motif, between E1768-P1777 (Seo et al, 2014; Chinen et al, 2020; Okumura et al, 2018). Further, NuMA contains a consensus SIM motif, 'IINI', (residues 1814–1817), that may encourage SUMO-SIM self-assembly. However, the mechanism by which NuMA SUMO1ylation supports spindle assembly, and why SUMO2 cannot substitute, is unexplored. While phase-separation concentrates proteins for mitotic spindle assembly (Sun et al, 2021), SUMO1 has a weaker influence on condensates than SUMO2/3, so further mechanistic analysis is required to explain why the SUMO1 conjugation is critical. Intriguingly, in WT cells, overexpression of SUMO proteins, particularly SUMO2, also results in multipolar spindles, suggesting that the disruption of variant balance can disrupt bipolar spindle development.

Finally, the first-in-class SUMOylation SUMO E1 inhibitor TAK-981 (Subasumstat) (Langston et al, 2021) is currently in clinical trials for solid tumours and is an exciting prospect for cancer treatment, particularly when coupled with immune-checkpoint inhibitors (Kukkula et al, 2021). Our findings provide a framework to explore whether the HDAC6–E1–SUMO1–NuMA axis contributes to the therapeutic efficacy or adverse effects (Dudek et al, 2021) associated with TAK-981 treatment.

# Methods

## Methods and protocols

### acK164-SAE2 antibody generation

Custom mouse monoclonal (clone 30E2-2) was raised against acetylated K164-SAE2 peptide (HP[Lys-Ac]PTQRTFPGC) by GenScript. Available on request to the corresponding author subject to completion of an M.T.A.

**Reagents and tools table**

| Reagent/resource | Reference or source | Identifier or catalog number |
|---|---|---|
| **Experimental models** | | |
| U2OS Flp-In TREx (Human Oesosarcoma, female) | Invitrogen | N/A |
| U2OS-EJ5-GFP | J. Stark Laboratory, City of Hope, Duarte, USA. | N/A |
| U2OS-EJ5-DR3 | J. Stark Laboratory, City of Hope, Duarte, USA. | N/A |
| Cell lines were verified mycoplasma-free and STR-tested. | | |
| **Recombinant DNA** | | |
| pcDNA5/FRT/TO-FLAG-SAE2-T2A-HA-SAE1 | GenScript | |
| pET28a-His-SAE1 | Addgene (Pichler et al, 2002) | #53135 |
| pET28b-His-SAE2 | Addgene (Werner et al, 2009) | #53117 |
| pET23a-Ubc9 | Addgene (Pichler et al, 2002) | #53137 |
| pcDNA5/FRT/TO-Myc-His-SUMO1/SUMO2 | GenScript | |
| pGEX4T-1-GST-SUMO1/SUMO2 | GenScript | |
| pET23a-His-RanGAP(aa 398-587) | Addgene (Flotho et al, 2012) | #53139 |
| pC3-GFP-NuMA(-K1766R) FL | C. Y. Choi (Sungkyunkwan University, Republic of Korea) | |
| pcDNA3-GFP-(SUMO1/SUMO2)-NuMA(-K1766R) Fragments | GenScript | |
| **Antibodies** | | |
| NUMA | Bio-techne | Cat# NBP2-54672, RRID AB_3339677 |
| β tubulin | Abcam | Cat# ab6046, RRID AB_2210370 |
| β-actin | Abcam | Cat# ab8227, RRID AB_2305186 |
| acK164-SAE2 (30E2-2) | GenScript | Custom design, this report. |
| SAE1 | Abcam | Cat# ab185552 |
| SAE2 (UBA2) | Sigma-Aldrich | Cat# HPA041436, RRID AB_2677479 |
| UBC9 | Abcam | Cat# ab75854, RRID AB_1310787 |

| Reagent/resource | Reference or source | Identifier or catalog number |
|---|---|---|
| SUMO1 (Y299) | Abcam | Cat# ab32058, RRID AB_778173 |
| SUMO1 (EP298) | Abcam | Cat# ab133352, RRID AB_11156108 |
| SUMO2/3 (8A2) | Abcam | Cat# ab81371, RRID AB_1658424 |
| His | Sigma-Aldrich | Cat# H1029, RRID AB_260015 |
| FLAG (M2) | Sigma-Aldrich | Cat# F1804, RRID AB_262044 |
| GFP | Roche | Cat# 11814460001, RRID AB_390913 |
| H2AX | Abcam | Cat# ab2893, RRID AB_303388 |
| αTubulin (DM1A) | Novus Biologicals | Cat# NB100-690, RRID AB_521686 |
| pS10-H3 | Invitrogen | Cat# MA5-15220, RRID AB_11008586 |
| pS10-H3 | Antibodies.com | Cat# A94899 |
| Donkey α Mouse AlexaFluor 488 | Life Technologies | Cat# A21202, RRID AB_141607 |
| Donkey α Rabbit AlexaFluor 488 | Life Technologies | Cat# A21206, RRID AB_2535792 |
| Donkey α Mouse AlexaFluor 555 | Life Technologies | Cat# A31570, RRID AB_2536180 |
| Donkey α Rabbit AlexaFluor 555 | Life Technologies | Cat# A31572, RRID AB_162543 |
| Donkey α rat AlexaFluor 555 | Life Technologies | Cat# A21434, RRID AB_2535855 |
| Rabbit α Mouse HRP | Dako | Cat# P0161, RRID AB_2687969 |
| Swine α Rabbit HRP | Dako | Cat# P0217, RRID AB_2728719 |
| Mouse TrueBlot® ULTRA: Anti-Mouse Ig HRP | Rockland | Cat# 18-8817-30, RRID AB_2610849 |
| **Oligonucleotides and other sequence-based reagents** | | |
| Cloning and mutagenesis primers: | This study (Custom, Merck Life Science, UK) | N/A |
| SAE1 Fwd Seq. (from 177 nt), GAAAGGACTGACCATGCTGG | " | |
| SAE2 Fwd Seq. (from 213 nt), GGCACAGGTTGCCAAGG | " | |
| SAE2 C173G_F, GAGAACCTTTCCTGGCGGTACAATTCGTAACAC | " | |
| SAE2 C173G_R, GTGTTACGAATTGTACCGCCAGGAAAGGTTCTC | " | |
| SAE2 K164Q_F, GTTATGAGTGTCATCCTCAGCCGACCCAGAGAAC | " | |
| SAE2 K164Q_R, GTTCTCTGGGTCGGCTGAGGATGACACTCATAAC | " | |
| SAE2 K164R_F, GTGTTATGAGTGTCATCCTAGGCCGACCCAGAGAACCTTTC | " | |
| SAE2 K164R_R, GAAAGGTTCTCTGGGTCGGCCTAGGATGACACTCATAACAC | " | |
| SAE2 S95N-M97T_F, GCCTACCATGACAACATCACGAACCCTGACTAT | " | |
| SAE2 S95N-M97T_R, ATAGTCAGGGTTCGTGATGTTGTCATGGTAGGC | " | |
| SUMO1 Fwd Seq., GGAGGCAAAACCTTCAACTG | " | |
| SUMO1 E93Q_Fwd, AGTTTATCAGCAACAAACGGG | " | |

| Reagent/resource | Reference or source | Identifier or catalog number |
|---|---|---|
| SUMO1 E93Q_Rev, CCCGTTTGTTGCTGATAAACT | " | |
| SUMO2 Fwd Seq., GAAAAGCCCAAGGAAGGAG | " | |
| SUMO2 Q89E_Fwd, GTGTTCCAAGAGCAGACGG | " | |
| SUMO2 Q89E_Rev, CCGTCTGCTCTTGGAACAC | " | |
| siRNA sequences: | | |
| siPIAS1 | Dharmacon | L-008167-00 |
| siUBC9 exon 8, AGCAGAGGCCUACACGAUUUA | Sigma-Aldrich (Garvin et al, 2022) | N/A |
| All other siRNA sequences | This study | |
| NTC (Renilla Luciferase), CUUACGCUGAGUACUUCGA | Sigma-Aldrich | N/A |
| SAE1 (Exon 4), GCAUGAGUUUGUAGAGGAGAA | Sigma-Aldrich | N/A |
| SAE2 (Exon 16), GCACCAGAUGUCCAAAUUGAA | Sigma-Aldrich | N/A |
| NuMA, GGCGUGGCAGGAGAAGUUCUU | Sigma-Aldrich | N/A |
| **Chemicals, enzymes and other reagents** | | |
| Nocodazole | Sigma | 487929 |
| Panobinostat | SignalChem | H83-904G |
| ACY-738 | MedChemExpress | HY-19327 |
| RGFP966 | SelleckChem | S7229 |
| A-485 | MedChemExpress | HY-107455 |
| NU9056 | Apexbio | A4492 |
| Remodelin Hydrobromide | Merck | 949912-58-7 |
| Butyrolactone 3 | Apexbio | C3209 |
| ML792 | Cambridge Bioscience | HY-108702 |
| **Software** | | |
| Las X | https://www.leica-microsystems.com/products/microscope-software/p/leica-las-x-ls/downloads/ | |
| GraphPad Prism 10 | https://www.graphpad.com/features | |
| Summit 6.2 | Beckman Coulter | |
| Thermo Scientific HCS Studio 4.0 Cell Analysis Software | https://www.thermofisher.com/uk/en/home/life-science/cell-analysis/cellular-imaging/high-content-screening/hcs-studio-2.html | |
| ImageJ | https://imagej.net/ij/download.html | |

New materials and reagents are available on request to the corresponding author.

### Generation of plasmids

The SAE2-T2A-SAE1 pcDNA5/FRT/TO construct was designed by AJG and generated by GenScript using KpnI and NotI restriction sites. pET28a-SAE1 was cloned using NheI and BamHI restriction sites; pET28b-SAE2 was cloned using NcoI and NheI; pET23a-UBC9 was cloned using NdeI and BamHI; and pET23a-RanGAP1 was made using NdeI and BamHI restriction sites. pGEX4T-1-SUMO1/2 were designed by AJG and made by GenScript by cloning SUMO1/2 cDNA into BamHI and EcoR1 restriction sites. pcDNA5/FRT/TO-Myc-His-SUMO1 and SUMO2 constructs were designed by AJG and generated by GenScript using BamHI and XhoI restriction sites. Full length NuMA constructs were generously gifted by Dr. Choi. AJL designed the cloning of the (SUMO1/2-)NuMA fragment cDNA into pcDNA3.1+N-eGFP at BamHI and XhoI, and the constructs were made by GenScript.

### Site-directed mutagenesis

Primers were designed for mutagenesis (Reagent and Tools Table), with mutagenesis performed by PCR using PfU (Promega). All mutagenesis was confirmed by Sanger Sequencing (Source Bioscience).

### Tissue culture

Parental FlpIn™ U2OS cells (Invitrogen) were cultured and grown in Dulbecco's Modified Eagle Media (DMEM) supplemented with 10% Fetal Calf Serum (FCS) and 1% Penicillin/Streptomycin. Cells were cultured in Corning T75 flasks and 10 cm² plates and kept at 37 °C and 5% $CO_2$. Once cells reached 70–80% confluency, they were passaged. Cells were tested for Mycoplasma using the LookOut Mycoplasma PCR detection kit (Merck).

### Inducible stable cell line generation

U2OS^TrEx-Flp-In™ were co-transfected with SAE2-T2A-SAE1 cDNA in the pcDNA5/FRT/TO vector and the Flp-recombinase cDNA in the pOG44 vector at a 6:1 pcDNA5/FRT/TO DNA: pOG44 DNA ratio using FuGene6 (Roche) at a ratio of 3.5:1 FuGene (μl): DNA (μg). Blank control transfections were performed as a control for selection. Two days after transfection, cells were selected with 150 μg/ml Hygromycin (Thermo Fisher Scientific) with culture medium replaced every 2–3 days; selection was exerted for ~2 weeks. After selection, cells were expanded and tested for expression of siRNA-resistant HA-SAE1/FLAG-SAE2 through treatment with siSAE1/siSAE2 (5 nM each) and 4 μg/ml Doxycycline for 72 h. Cell lysates were prepared in 4×SDS loading buffer and western blot analysis performed.

### Plasmid and siRNA transfection

FuGene6 (Roche) was used at 2:1 FuGene (μl): DNA (μg), following the manufacturer's guidelines. SUMO2 and SUMO1 overexpression was achieved using 0.5 μg of DNA per well of a 24-well plate for the durations indicated in the figure. GFP-NuMA constructs were transfected at 1 μg/ml. siRNA was introduced to cells using the transfection reagent Dharmafect1 (Dharmacon) following the manufacturer's instructions, from working concentrations of 10 nM (NTC, UBC9, PIAS1, NuMA) and 5 nM (SAE1, SAE2).

### FLAG immunoprecipitation

U2OS were cultured at 37 °C/5% $CO_2$ in 15 cm² dishes supplemented with 4 μg/ml Doxycycline (Sigma) for 48 h to induce exogenous SAE1:SAE2 expression. Cells washed with 1 ml ice-cold TBS (20 mM Tris/HCl pH 7.5, 150 mM NaCl) before suspension in RIPA Buffer (50 mM Tris-HCl pH 7.5, 150 mM NaCl, 1% TritonX100, 0.25% sodium deoxycholate, 0.1% SDS, 1 mM EDTA, 10 mM NaF) plus EDTA-free protease inhibitor cocktail (Roche) and PhosSTOP (Roche). Lysis mix was incubated on ice for 10 min and sonicated at 50% intensity for 10 s. Samples were centrifuged at $14,000 \times g$/4 °C for 10 min, and supernatant combined with 15 μl FLAG(M2) agarose (Sigma), incubated with mixing O/N at 4 °C. Beads were pelleted by centrifugation at $1000 \times g$/4 °C for 2 min. Supernatant was discarded and beads washed with TBST. 30 μl 4×SDS loading buffer was added to beads with boiling at 95 °C for 10 min and centrifuged at $5000 \times g$/RT to pellet beads. Immunoprecipitations using histone acetyltransferase (HAT) inhibitors were conducted as above; however, before lysing, cells were incubated with 2.5 μM each of indicated inhibitor against p300 (A-485), TIP60 (NU9056), NAT10 (Remodelin hydrobromide), and GCN5 (Butyrolactone 3). To immunoprecipitate material from mitotic cells, 100 ng/ml of Nocodazole was added for a total of 17 h to relevant dishes, 48 h after SAE1:SAE2 induction. HDAC inhibitors (Panobinostat [Broad spectrum HDAC inhibitor], ACY-738 [HDAC6 inhibitor], RGFP966 [HDAC3 inhibitor]) were applied at 2.5 μM for 2 h prior to the release from Nocodazole treatment. At 17 h, cells were washed twice with PBS and released into mitosis for 10 min in media supplemented with relevant HDAC inhibitor. Cells were then harvested via mitotic shake off and lysed as above in ice-cold RIPA Buffer (50 mM Tris-HCl pH 7.5, 150 mM NaCl, 1% TritonX100, 0.25% Sodium deoxycholate, 0.1% SDS, 1 mM EDTA, 10 mM NaF) plus EDTA-free protease inhibitor cocktail (Roche) and PhosSTOP (Roche) supplemented with 2 μM Panobinostat.

### GFP-trap immunoprecipitation

GFP-nanobody sepharose beads were prepared and stored in 70% ethanol at 4 °C. U2OS were transiently transfected with 20 μg pcDNA3-GFP-NuMA^1708-1982 and treated with 100 ng/ml nocodazole (16 h) and mitotic cells were harvested by mitotic shake off. Mitotic U2OS were lysed in cold lysis buffer (10 mM HEPES-pH 7.6, 200 mM NaCl, 1.5 mM $MgCl_2$, 10% glycerol, 0.2 mM EDTA, 1% Triton, cOmplete protease inhibitor and PhosSTOP) and sonicated and then centrifuged at $14,000 \times g$ for 5 min and the supernatant was separated. For each condition, 20 ul GFP-trap beads were washed twice in TBS and once with buffer and then incubated with cell lysates for overnight at 4 °C with agitation. GFP-trap:cell lysates were centrifuged at $1000 \times g$ 4 °C for 2 min and the supernatant was discarded, and GFP-trap beads were washed three times in buffer. Finally, GFP-trap beads were treated with 4×loading buffer and boiled prior to western blot analysis.

### Western blotting

For a list of antibodies, see Reagent and Tools Table and Appendix Table S1. Protein samples in loading buffer were subject to SDS-PAGE and transferred onto Immobilon-P PVDF-membrane (Merck). Membranes were blocked in 5% milk in PBST or in 5% BSA with TBST for 30 min. Incubation with primary antibodies for 16 h/4 °C/rolling. Membranes washed in PBST/TBST for $3 \times 10$ min and then incubated with relevant secondary HRP antibodies in blocking solution for minimum 1 h/RT. Membranes washed in PBST/TBST for $3 \times 10$ min and HRP stimulated with EZ-ECL mix (Biological Industries) or ECL Prime (Amersham). Blots exposed to X-ray film (Wolflabs) and developed with KONICA MINOLTA SRX-101A. Densitometry calculations performed using ImageJ.

### Protein overexpression and purification

BL21 (DE3; NEB) were transformed with the relevant plasmids. Starter cultures were established by inoculating 40 ml LB (Melford; kanamycin 50 μg/ml or ampicillin 100 μg/ml) with a single colony, grown O/N at 37 °C. 10 ml starter culture was used to inoculate each litre LB (kanamycin 50 μg/ml or ampicillin 100 μg/ml) and grown at 37 °C/180 rpm to OD$_{595}$ ~0.6. Protein overexpression induced with IPTG and temperature adjusted as follows, SUMO1 and SUMO2 at 0.5 mM IPTG/18 °C/18 h; SAE1 and SAE2 at 1 mM IPTG/25 °C/6 h; UBC9 at 1 mM IPTG/37 °C/4 h; RanGAP1 (aa 398–587) at 1 mM IPTG/37 °C/4 h with shaking at 180 rpm. Overexpression protocols for SAE1:SAE2, UBC9, and RanGAP1 (aa 398–587) were adapted from Flotho et al, 2012 (Flotho and Werner, 2012). BL21(DE3) cells were harvested by centrifugation at $5000 \times g$/4 °C/10 min with the resulting pellet resuspended in 10 ml cold lysis buffer (20 mM Tris-HCl pH 8, 130 mM NaCl, 1 mM EGTA, 1 mM EDTA, 1% Tritonx100, 10% Glycerol, 1 mM DTT, EDTA-free protease inhibitor (Roche)). Separately overexpressed SAE1 and SAE2 combined here. 0.5 mg/ml lysozyme was added and incubated for 30 min/4 °C/rolling. 1 U/ml DNase (Thermo Fisher) added before sonication at $5 \times 30$ s at 100% intensity with 2-min recovery—all on ice. Samples were centrifuged at $48,000 \times g$/4 °C/30 min in a JLA-25.50 and supernatant was filtered through a 0.45 μm PES membrane (Millex) and combined protein-tag-specific resin. Prior to use, resins were washed twice in PBS and once in the lysis buffer, with centrifugation performed at $1000 \times g$/4 °C/3 min. Respective protein purification continued as follows.

### SUMO1 and SUMO2 purification

GST-SUMO1/SUMO2 were combined with 250 µl glutathione Sepharose 4B beads (Cytiva) and incubated for 3 h/4 °C/rolling. Beads were centrifuged at $1000 \times g$/4 °C/10 min with supernatant collected. Wash steps comprised $3 \times 10$ ml lysis buffer suspension of beads and $1 \times 10$ ml cleavage buffer (20 mM Tris-HCl pH 8.4, 150 mM NaCl, 1.5 mM $CaCl_2$) with centrifugation as above. Beads were then suspended in 500 µl cleavage buffer supplemented with 16 U Thrombin cleaving protease and incubated at 4 °C/16 h/rolling. Samples were then centrifuged at $1000 \times g$/4 °C/3 min and the supernatant was collected for centrifugation at $14{,}000 \times g$/4 °C/20 min to clear any beads or aggregate. The 500 µl sample was subjected to size-exclusion chromatography (SEC) through an AKTA pure™ (UNICORN™ software) Superdex200 Increase 10/300 GL column equilibrated in 20 mM HEPES pH 7.5, 100 mM NaCl, 0.5 mM TCEP: 0.5 ml fractions collected. Fractions constituting a $UV_{280}$ trace peak were analysed by SDS-PAGE stained with InstantBlue (Lubioscience). Pure protein fractions were pooled and stored at $-80$ °C at 1 mg/ml.

### UBC9 purification

Column filled with 10 ml SP-Sepharose beads and 60 ml UBC9 lysate applied at ~1 ml/min; FT collected and reapplied. UBC9 lysis buffer was passed through a column for wash step. 20 ml UBC9 elution buffer (50 mM Na-phosphate pH 6.5, 300 mM NaCl, 1 mM DTT, 1 cOmplete protease inhibitor (EDTA-free) tablet/50 ml) applied to beads and 1.5 ml fractions collected—10 µl samples analysed by 15% SDS-PAGE and InstantBlue stain. Fractions with the greatest quantity and purity of UBC9 protein combined and concentrated down to 5 ml using 3-kDa MWCO centrifugal concentrator (Thermo Scientific) at $4000 \times g$/4 °C. Sample cleared by centrifugation at $14{,}000 \times g$/4 °C/20 min before SEC through a Superdex75 equilibrated in transport buffer (20 mM Hepes pH 7.3, 110 mM potassium acetate, 1 mM EGTA, 1 mM DTT, 1 cOmplete protease inhibitor tablet/L): 4 ml fractions collected. Fractions constituting $UV_{240}$ peak were analysed by 15% SDS-PAGE and Instantblue stain. Pure UBC9 protein fractions were pooled and concentrated as before, and finally aliquoted and stored at $-80$ °C at 8 mg/ml.

### SAE1:SAE2 and RanGAP1 (aa 398–587) purification

These His-tagged protein lysates were combined with 1 ml nickel beads (Sigma) and incubated at 4 °C/2 h/rolling. Samples were then centrifuged at $1000 \times g$/4 °C/10 min to pellet nickel beads; FT collected and retained. Nickel-bead:His-protein pellet suspended in 10 ml wash buffer ahead of centrifugation as before; supernatant retained. Nickel beads resuspended in 5 ml elution buffer and centrifuged as before; supernatant extracted and pushed through 0.45-µm PES filter. This protein suspension was run using an AKTA pure™ (UNICORN™ software) on a HiLoad 16/600 Superdex200 pg column equilibrated with 20 mM Hepes pH 7.5, 100 mM NaCl, 0.5 mM TCEP buffer: 2 ml fractions collected. Fractions corresponding with a $UV_{280}$ peak were analysed by SDS-PAGE stained with InstantBlue. Fractions containing the purest SAE1:SAE2 or RanGAP1 (aa 398–587) were pooled and exposed to $4000 \times g$ at 4 °C in centrifugal concentrators with 30 kDa and 10 kDa MWCO, respectively. Proteins were aliquoted and stored at $-80$ °C at 1 mg/ml (SAE1:SAE2) and 21 mg/ml (RanGAP1).

### Microscale thermophoresis (MST)

SAE1:SAE2 was labelled using Protein Labelling Kit RED-NHS 2nd Generation (NanoTemper Technologies). The labelling reaction was performed according to the manufacturer's instructions in the supplied labelling buffer, using 20 µM SAE1:SAE2 and a molar dye: protein ratio $\approx 3{:}1$ at RT for 30 min in the dark. Unreacted dye was removed with the supplied dye removal column equilibrated with MST buffer (20 mM Hepes pH 8.35, 150 mM NaCl, 0.5 mM TCEP). The degree of labelling was determined using UV/VIS spectrophotometry at 650 and 280 nm. A degree of labelling of 0.5–0.6 was typically achieved.

The labelled SAE1:SAE2 protein was adjusted to 20 nM with MST buffer supplemented with 0.005% Tween20. The SUMO1/SUMO2 ligand was dissolved in MST buffer supplemented with 0.005% Tween20, and a series of 16 1:1 dilutions was prepared using the same buffer. For the measurement, each ligand dilution was mixed with one volume of labelled SAE1:SAE2 protein for a final concentration of 10 nM and ligand concentrations ranging from 125 mM to 0.00381 µM, respectively. After 10 min incubation, samples were centrifuged at $10{,}000 \times g$ for 10 min, and loaded into Monolith NT.115 [Premium] Capillaries (NanoTemper Technologies). MST was measured using a Monolith NT.115 [NT.115Pico/N.T.LabelFree] instrument (NanoTemper Technologies) at an ambient temperature of 22 °C. Instrument parameters were adjusted to 20% LED power and high [low/medium] MST power. Data of three independently pipetted measurements were analysed (MO.Affinity Analysis software version 2.3, NanoTemper Technologies) using the signal from an MST-on time of 2.5 s.

### SUMO adenylation assay

Reactions were performed by combining 30 µM SAE1:SAE2-K164 variants with 40 µM SUMO, and 1 U pyrophosphatase. Reactions were initiated by adding 150 µM BODIPY-ATP and incubated at 30 °C for 10 min. Reactions were quenched by the addition of 4×loading buffer and incubated at 95 °C for 5 min before running samples on 15% SDS-PAGE. Gels were imaged using excitation at 488 nm to excite BODIPY-ATP and bands at 15 kDa were taken to be SUMO1-AMP-BODIPY.

### SAE1:SAE2-loading assays

Conducted in $V_T$ 20 µl in 20 mM HEPES pH 7.5, 50 mM NaCl, 5 mM $MgCl_2$ with 5 µM SAE1:SAE2 and 5 mM ATP with reactions started by adding 10 µM SUMO1-C52A-S9C-Alexa488 or SUMO2-C48A-A2C-Alexa647 and incubating samples on ice for 15 s. The reaction was terminated with 20 µl 4×Loading buffer (reducing-agent free) and incubated at 95 °C. Samples were processed for analysis by SDS-PAGE and imaged at excitation wavelengths of 488 nm and 647 nm, respectively. The band at 120 kDa was taken as SAE2~SUMO product.

### SUMOylation assays

All proteins were diluted in SUMO assay buffer (SAB; 20 mM HEPES pH 7.5, 50 mM NaCl, 5 mM $MgCl_2$, 0.1 mM DTT) and quantified on a NanoDrop2000/2000c (Thermo Fisher Scientific), using SAB to adjust proteins to concentrations of 0.2 mg/ml SAE1:SAE2 ($\varepsilon$/1000 = 69.550, 109.655 kDa), 0.15 mg/ml UBC9 ($\varepsilon$/1000 = 29.700, 18.007 kDa), 4.5 mg/ml RanGAP1(aa 398-587) ($\varepsilon$/1000 = 10.805, 22.386 kDa), 2 mg/ml, SUMO1 ($\varepsilon$/1000 = 4.470, 11.277 kDa), 2 mg/ml SUMO2 ($\varepsilon$/1000 = 1.490, 10.753 kDa).

Reaction mixes prepared to $V_T$ 20 μl SAB with 25 nM SAE1:SAE2, 100 nM UBC9, 10 μM RanGAP1 (aa 398–587), 10 μM SUMO1, and 10 μM SUMO2. The reaction was started by adding 5 mM ATP with incubation at 30 °C for 10 min, and reactions were terminated by adding 20 μl 4 × Loading buffer (reducing-agent free) and incubating samples at 95 °C for 10 min. Samples were centrifuged at 14,000 × *g* for 5 min before processing by SDS-PAGE and western blot analysis.

### Densitometry

Densitometry was calculated using ImageJ (Rueden et al, 2017) to quantify western blot band intensities. All quantification is from at least three independent experiments. To quantify RanGAP1-SUMO from in vitro assay western blots, band intensities were measured, and the background was subtracted. Values were normalised against the WT SAE1:SAE2:SUMO1/SUMO2-only condition RanGAP1-SUMO product intensity. For the densitometry to calculate the levels of SUMO conjugation, we calculated the relative amounts of SUMO 'smear' in cells using densitometry with ImageJ. The amount was then normalised to a GAPDH loading control.

### Immunofluorescent staining

The staining for γH2AX foci kinetics was performed as follows. Cells were plated directly onto 48-well plates at $1 \times 10^4$ cells/ml and allowed to settle overnight. siRNA was applied to cells for 72 h and complemented with SAE1:SAE2 variants by the addition of 4 μg/ml Doxycycline. Prior to 2 Gy irradiation cells were pulsed with 1 μM EdU for 10 min. 0 Gy samples were fixed directly after Edu incubation. After irradiation, cells were allowed to recover for allotted timepoints. At allotted timepoints cells were pre-extracted using CSK buffer (100 mM NaCl, 300 mM sucrose, 3 mM MgCl₂, 10 mM PIPES pH 6.8, 0.7% Triton x100) for 1 min RT before fixation with 4% PFA in PBS. Fixed cells were permeabilised for a further 5 min using 0.5% TritonX100 in PBS before incubation with blocking solution (10% FCS in PBST) for 30 min. EdU was labelled by Click-iT® chemistry according to the manufacturer's protocols (Life Technologies) with Alexa-647-azide. Cells were washed then blocked for a further 30 min before incubation with primary antibody diluted in blocking solution for 1 h RT. See Reagent and Tools Table for list of antibodies and Appendix Table S1 for dilutions. Following this the samples were washed in PBST before incubation with the fluorescent secondary antibody for 1 h RT. Samples were washed three times in PBS before the DNA was stained using Hoechst at a 1:50,000 concentration for 5 min. Excess of Hoechst was washed with PBS before antibodies were fixed in place for 5 min using 4% PFA. PBS was reapplied to cells and imaging proceeded within 3 days.

Cells for micronuclei and mitotic spindle staining were plated at $2 \times 10^4$ cells/ml in 24-well plates on glass coverslips and treated with siRNA and Doxycycline as described above. Micronuclei samples were fixed 72 h later with 4% PFA before immunofluorescent staining.

Mitotic spindle samples were all treated with 100 ng/ml nocodazole for 16 h. Cells were then washed twice in PBS before media was replaced to allow mitosis to progress for 35 min before fixation with PFA. Samples using siRNA-resistant constructs were treated with siRNA and Doxycycline for 48 h prior to the addition of nocodazole and cells were released into mitosis in media supplemented with Doxycycline. Deviations and additions to this basic protocol such as the use of inhibitors and or additional transfections are outlined schematically in the relevant figure.

Immunostaining of cells for micronuclei and mitotic spindle apparatus proceeded as follows: Cells were permeabilised for 5 min using 0.5% TritonX100 in PBS and blocked in blocking solution for 1 h RT before the addition of primary antibodies. See Reagent and Tools Table for list of antibodies and Appendix Table S1 for dilutions. Samples were washed in PBST before the addition of secondary antibodies. DNA was stained with Hoechst and coverslips were mounted onto glass slides.

### Microscopy and analysis

High content γH2AX foci imaging and analysis was conducted using the CellInsight CX5 HCS Platform with ×20 objective lens using Compartmental Analysis BioApplication software. Spot detection within the nuclear compartment was used to count foci. Raw data outputs from this analysis were extracted so that cell cycle positioning could be achieved by plotting the total nuclear intensity of the Hoechst signal against the log average nuclear intensity of the EdU signal. This was used to separate γH2AX foci numbers into S phase, G1 and G2 populations. All other staining was imaged using the Leica DM6000B microscope using a 40x objective and HBO lamp with 100 W mercury short arc UV bulb light source and four filter cubes, A4, L5, N3 and Y5, which produce excitations at wavelengths 360, 488, 555, and 647 nm, respectively. To analyse micronuclei, all cells were counted in at least 4 fields of view to reduce sampling bias, and a minimum of 400 cells were counted per condition per experimental repeat. Mitotic spindles were assessed by counting all laterally presented metaphase and anaphase cells in a field of view, with a minimum of 50 cells counted per condition per experimental repeat. All cells in a field of view were counted from least 4 fields of view. Metaphase and anaphase cells were identified based on the well characterised morphology that tubulin and DNA adopt during these mitotic stages, pericentrin staining was used as a further aid this determination when a spare channel for imaging allowed for its inclusion.

### Denaturing SUMO immunoprecipitation on mitotic cells

The methodology for denaturing SUMO immunoprecipitations was adapted from (Becker et al, 2013). In brief, 8× 10 cm dishes were plated per condition. At 80% confluency cells were treated with 100 ng/ml nocodazole, 1 μM ML792 and 4 μg/ml Doxycycline for 16 h. Cells were washed twice in PBS, released into growth media supplemented with ML792 and Doxycycline for 10 min before harvesting via mitotic shake off. The resulting pellets were lysed in 200 μl 1% SDS lysis buffer (20 mM sodium phosphate, pH 7.4, 150 mM NaCl, 1% SDS, 1% Triton, 0.5% sodium deoxycholate, 5 mM EDTA, 5 mM EGTA, 10 mM NEM, plus EDTA-free protease inhibitor cocktail (Roche) and PhosSTOP (Roche) and sonicated until the viscous sample became fluid. Samples were boiled for 10 min with 50 mM DTT and centrifuged at 14,000 × *g* for 15 min. 30 μl of supernatant was taken and combined with 30 μl of 4×SDS loading buffer for use as an input on western blots. The remaining supernatant was diluted 1:10 in RIPA without SDS (20 mM sodium phosphate, pH 7.4, 150 mM NaCl, 1% Triton, 0.5% sodium deoxycholate, 5 mM EDTA, 5 mM EGTA, 10 mM NEM, plus EDTA-free protease inhibitor cocktail (Roche) and PhosSTOP (Roche). Total protein was quantified for each sample using Pierce™ 660 nm Protein Assay Reagent following manufacturer guidelines. The lysates were equalised for protein content and split so that half was used for SUMO1, and half was used for SUMO2/3 immunoprecipitations. Pierce™ Protein A/G Agarose beads

(Thermo Fisher) were washed 3× in 0.1% SDS RIPA (20 mM sodium phosphate, pH 7.4, 150 mM NaCl, 0.1% SDS, 1% Triton, 0.5% sodium deoxycholate, 5 mM EDTA, 5 mM EGTA) before rotating in 0.1% RIPA with primary antibody at room temperature for 1 h. Thirty microliters of beads were prepared for each sample using 3 µg of antibody per 30 µl of beads, ab32058 (Abcam) was used for SUMO1 and ab81371 (Abcam) was used for SUMO2/3 immunoprecipitations. Antibody-bound beads were pelleted and combined with prepared lysates prior to rotation at 4 °C O/N. Beads were pelleted and washed 3 × 3 mins with agitation in 0.1% RIPA before protein elution at 95 °C in 4xSDS loading buffer.

### Colony survival assays

U2OS Flp-In™ cells were plated in 24-well plates at $2 \times 10^4$/ml and treated with siSAE1:siSAE2 (5 nM each) and Doxycycline for 72 h. Cells were heat shocked in a water bath at 43 °C for 40 min with control condition placed in a 37 °C water bath. Cells were suspended in 100 µl 1×Trypsin followed by 900 µl DMEM and plated at limiting dilutions in 6-well plates. Plates were incubated for 7 days at 37 °C and 5% $CO_2$, then stained with 0.5% crystal violet (50% methanol) and counting.

### DR-GFP and NHEJ-EJ5

U2OS-DR3-GFP and NHEJ-EJ5 (reporter cell lines) were a generous gift from Jeremy Stark (City of Hope, Duarte, USA). U2OS reporter cell lines were simultaneously co-transfected with siRNA using Dharmafect1 (Dharmacon) and DNA (RFP and *I-Sce1* endonuclease expression constructs) using FuGene6 (Promega), respectively. After 16 h, the media was replaced, and cells were grown for a further 48-h before fixation in 2% PFA in PBS. RFP and GFP double-positive cells were scored by FACS analysis using a CyAn flow cytometer. Ten thousand cells were counted per sample. Data was analysed using Summit 6.2 software. The percentage of GFP-positive cells was determined as a fraction of RFP positive cells (RFP only +double GFP-RFP) to control for transfection efficiency.

### Blinding

Blinding of samples to researchers was not carried out for cost reasons. Positive and negative controls were assessed prior to experimental samples to ensure efficient use of time.

### Statistics and reproducibility

Shapiro–Wilk test was used to assess the normality of the data. Normally distributed the parametric two-sided unpaired Student's *t* test, was used to detect the statistical differences between two groups and parametric one-factor analysis of variance (ANOVA) was used to detect the statistical differences among >2 groups. The *n* value is reported for each experiment.

## Data availability

This study includes no data deposited in external repositories. Raw Data for the main figures has been uploaded with the submission.

The source data of this paper are collected in the following database record: biostudies:S-SCDT-10_1038-S44318-025-00532-y.

## Peer review information

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

## Acknowledgements

Grant funding. Wellcome Trust 206343/Z/17/Z (MJ, AW, and AJG), Medical Research Council MR/X001008/1 (AL and MJ), University of Birmingham (AL). Biotechnology and Biological Sciences Research Council: BB/V01983X/1, BB/S017283/1, BB/P009840/1 (TJK) and Midlands Integrative Biosciences Training Partnership, BB/M01116X/1 (PW and BFC). Cancer Research UK: C8820/A28283 (MM). We thank Jeremy Stark (City of Hope, Duarte USA) for U2OS DR-GFP and NHEJ-EJ5 cells and Cheol Yong Choi (Sungkyunkwan University, Republic of Korea) for pC3-GFP-NuMA constructs. pET28a-His-SAE1, pET28b-His-SAE2, pET23a-His-RanGAP (aa 398–587) and pET23a-Ubc9 were gifts from Frauke Melchior (Addgene plasmids #53135, #53117, #53139 & #53137). We thank Helen Walden (University of Glasgow, UK) and Chris Lima (Sloan Kettering Institute, U.S.A) for their invaluable discussions about the project. We also thank the Microscopy and Imaging Services at Birmingham University (MISBU) and the UoB Flow Cytometry Services (UoBFC) in the Tech Hub facility for microscope and FACS support and maintenance. Model schematic in Fig. 5 created in BioRender. Lanz A (2025) https://BioRender.com/s07y449.

## Author contributions

**Alexander J Lanz**: Conceptualization; Formal analysis; Investigation; Visualization; Methodology; Writing—original draft; Writing—review and editing. **Alexandra K Walker**: Conceptualization; Formal analysis; Investigation; Methodology; Writing—original draft; Writing—review and editing. **Mohammed Jamshad**: Data curation; Formal analysis; Investigation; Methodology. **Alexander J Garvin**: Conceptualization. **Matthew Stewart**: Resources. **Peter Wotherspoon**: Data curation. **Benjamin F Cooper**: Data curation. **Matthew Mackintosh**: Resources. **Oliver Crutchley**: Investigation. **Timothy J Knowles**: Supervision. **Joanna R Morris**: Conceptualization; Supervision; Funding acquisition; Visualization; Writing—original draft; Project administration; Writing—review and editing.

Source data underlying figure panels in this paper may have individual authorship assigned. Where available, figure panel/source data authorship is listed in the following database record: biostudies:S-SCDT-10_1038-S44318-025-00532-y.

## Disclosure and competing interests statement

The authors declare no competing interests.

# Expanded View Figures

**A**

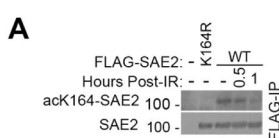

**B**

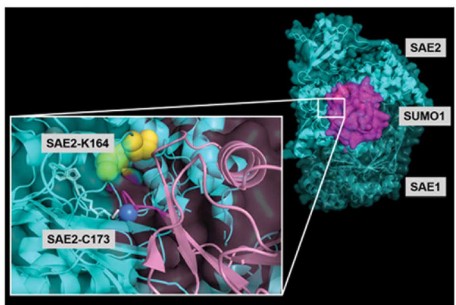

**Figure EV1.  Detection of acK164-SAE2 and location of SAE2-K164**

(A) Western blot analysis of U2OS cells, untransfected (-) or expressing FLAG-SAE2 or FLAG-SAE2-K164R and treated with 10 Gy IR, then with half an hour (0.5) or 1 h recovery (1). Lysates were subjected to anti-FLAG immunoprecipitation and western blots were probed with acetyl-K164-SAE2 (mouse monoclonal) and anti-SAE2 antibodies. Performed once. (B) Structure of SAE1:SAE2:SUMO1 (PDB: 3KYD) adapted from Olsen et al, (2010) represented as a ribbon structure of SAE1:SAE2 in cyan and SUMO1 in magenta. The magnified image shows the C-terminal tail of SUMO1 (dark magenta) extending toward SAE2-C173 (dark blue sphere) through a channel in SAE1:SAE2; the ceiling of the channel is in part formed by SAE2-K164 (yellow spheres, visable as green where through the cyan).

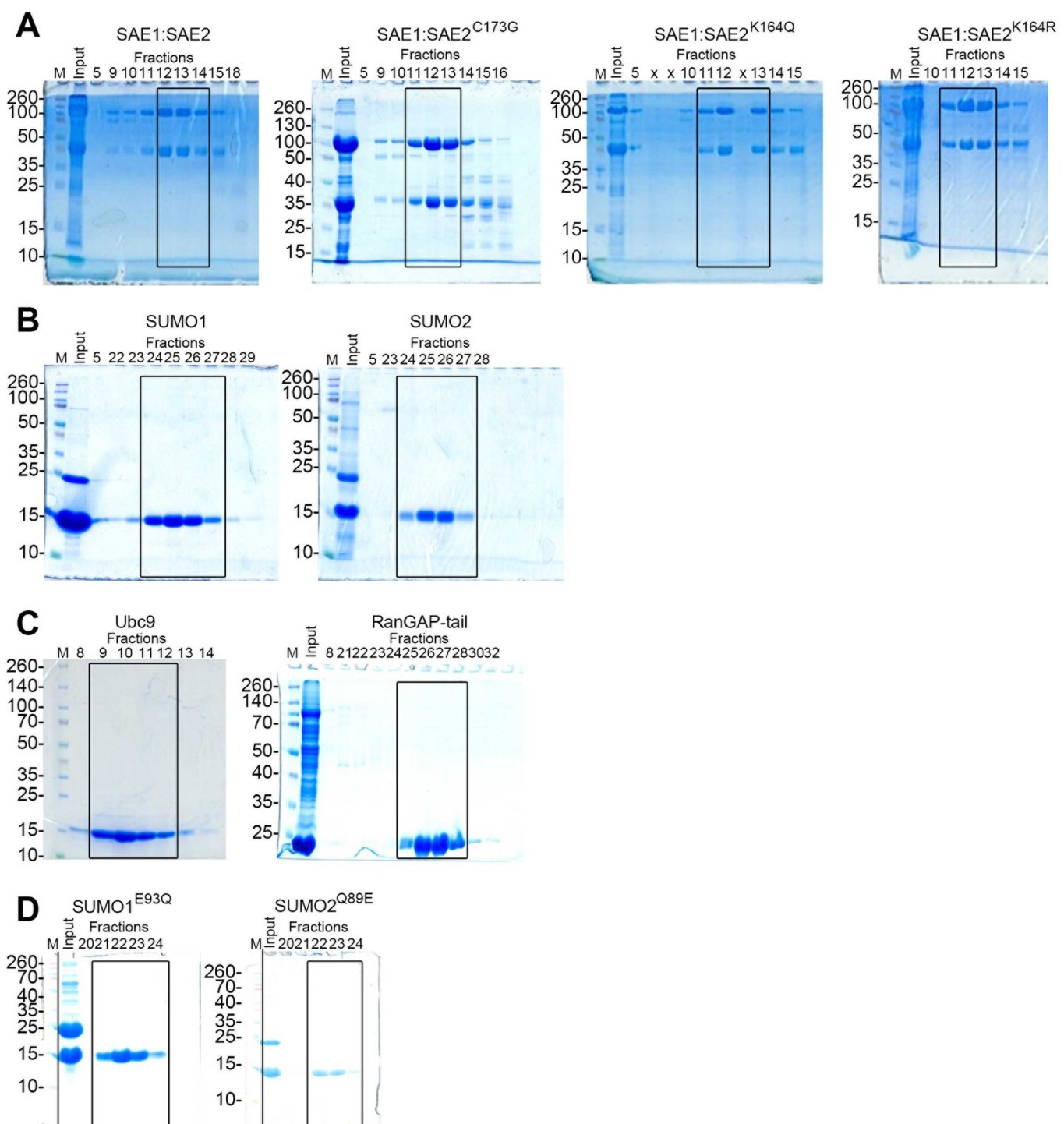

**Figure EV2. Coomassie gels for each recombinant protein prepared.**

Representative Instantblue stained SDS-PAGE gels from SEC fractions for the respective purified proteins. Shown here are gels from (**A**) SAE1:SAE2, SAE1:SAE2-C173G, SAE1:SAE2-K164Q, and SAE1:SAE2-K164R; (**B**) SUMO1 and SUMO2; (**C**) UBC9 and RanGAP1 (aa 398-587); and (**D**) SUMO1-E93Q and SUMO2-Q89E protein purifications.

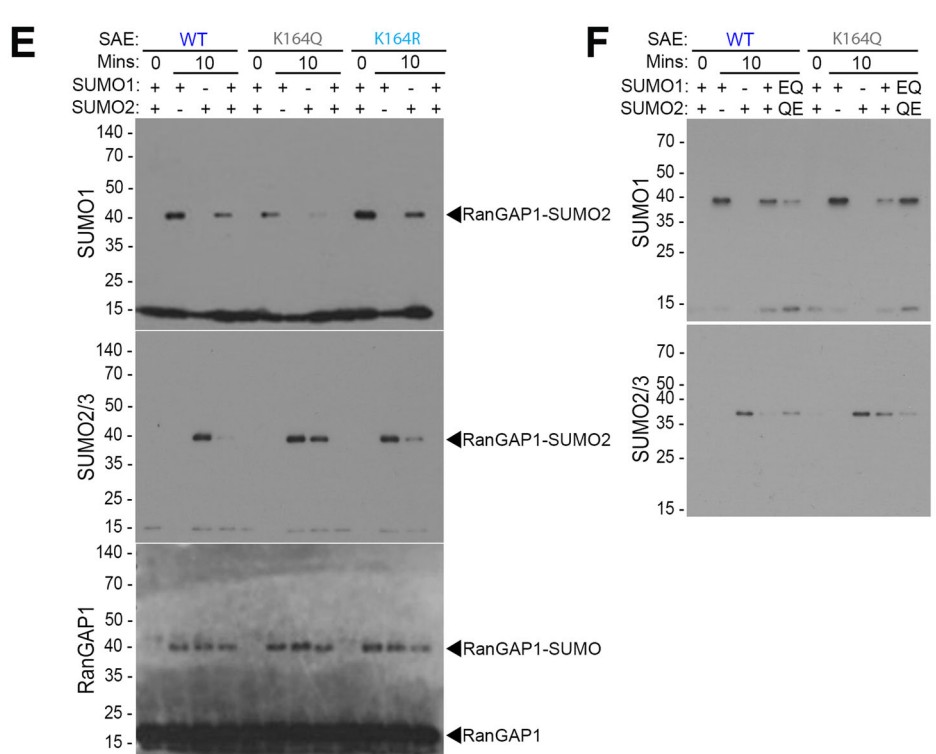

◀ **Figure EV3. Extended in vitro data.**

(A) MST Thermographs of SUMO1 or SUMO2 binding to SAE1:SAE2 or SAE1:SAE2-K164Q provide well-defined curves. The cold region is set to 0 s (blue) and the hot region set to 2.5 s (red) to determine the $K_d$ of the interaction and to avoid any potential convection phenomena. (B) SDS-PAGE gels from in vitro adenylation assays after combining 30 µM SAE1:SAE2 variants, 40 µM SUMO1 (top) or SUMO2 (bottom), and 150 µM BODIPY-ATP. Gels were imaged with excitation at 488 nm to observe the BODIPY-ATP. Gels were subsequently stained using SYPRO Ruby to check protein loading. Replicated 3-times in the laboratory and quantified in Fig. 2B. (C) Confirmation of thioester bond formation between SAE2 and Alexaflour-tagged SUMO proteins. Reactions comprised 1 µM SUMO1-C52A-S9C-Alexa488(left) or 1 µM SUMO2-C48A-A2C-Alexa647 (right), 200 nM SAE1:SAE2, and 5 mM ATP, incubated at 30 °C for 10 min. Reactions from the left, in lanes 1 and 4 lacked ATP, and lanes 3 and 6 were followed by 30 °C, 10 min incubation at 30 °C with 100 mM DTT to assess SAE2-SUMO thioester formation. The SUMO1-C52A-S9C-Alexa488 and SUMO2-C48A-A2C-Alexa647 loading were observed with excitation wavelengths of 493 nm and 647 nm, respectively, with bands at ~120 kDa taken to be SAE2~SUMO. Replicated once in the laboratory. (D) Representative SDS-PAGE gels for in vitro SUMO loading assays combining 10 µM SUMO with 5 µM SAE1:SAE2-K164 variants or SAE1:SAE2-C173G. Reactions were initiated by the addition of 5 mM ATP on ice for 15 s and terminated by the addition of reducing agent-free loading buffer and boiling samples. 10 µM SUMO1-C52A-S9C-Alexa488 and SUMO2-C48A-A2C-Alexa647 were observed with excitation wavelengths of 493 nm and 647 nm, respectively, with bands at 120 kDa taken to be SAE2~SUMO. SDS-PAGE gels were stained with SYPRO ruby to detect unconjugated SUMO1 (15 kDa), SAE1 (40 kDa), and SAE2 (100 kDa). The panel below shows the over-exposed image to show SUMO2 loading. Replicated 3-times in the laboratory and quantified in Fig. 2C. (E) Representative blots for the in vitro SUMOylation data in Fig. 2D. Reactions comprised 10 µM SUMO1 and/or 10 µM SUMO2, 25 nM SAE1:SAE2, 100 nM UBC9, 10 µM RanGAP1 (aa 398–587), and 5 mM ATP, incubated at 30 °C for 10 min. Conditions were processed by SDS-PAGE in duplicate, such that western blots were developed using αSUMO1 and αSUMO2/3 antibodies. Note that the bands indicated are those quantified. Replicated 3 times in the laboratory and quantified in Fig. 2D. (F) Representative blots for the in vitro SUMOylation data in setup as described for EV3D and including a condition with 20 µM SUMO1-E93Q and SUMO2-Q89E. As before, samples were processed by SDS-PAGE in duplicate and western blots were developed using αSUMO1 and αSUMO2/3 antibodies. Replicated 3-times in the laboratory and quantified in Fig. 2F.

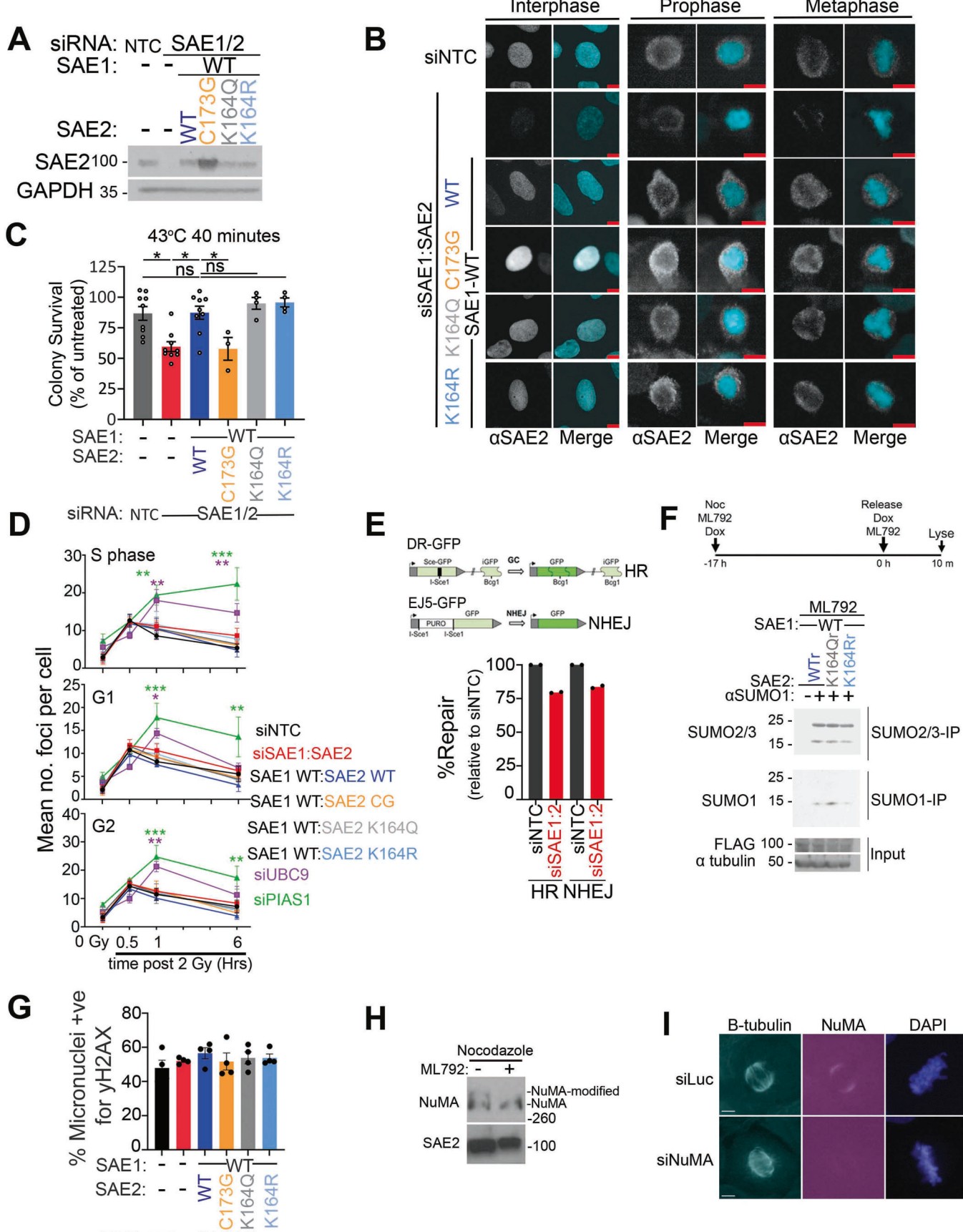

◀

**Figure EV4.  Extended assessment of the cellular impacts of SAE2 variants.**

(A) Representative western blot of SAE2 expression in stable, inducible siRNA-resistant SAE1:SAE2 variant U2OS cells. Cells were treated for 72 h with siRNA for either NTC or SAE1:SAE2 with the concurrent addition of 4 μg/ml Doxycycline to induce expression of the indicated integrated SAE1:2 constructs. Replicated >5 times in the laboratory. (B) Representative images depicting the localisation of SAE2 in interphase, prophase and metaphase U2OS cells. Cells depleted for SAE1:SAE2 and complemented with WT or indicated SAE1:SAE2-variants. Interphase cells received no synchronisation. Prophase and metaphase cells were synchronised in nocodazole for 16 h and either fixed immediately or after a 35 min release into mitosis, respectively. Five-micrometer scale bar is shown as a red line. Cells chosen from a representative field >20 similar cells. Performed once. (C) U2OS depleted for SAE1:SAE2 and complemented with WT or indicated SAE1:SAE2-variants, subjected to 43 °C for 40 min before replating and counting after colony growth. Significance calculated using one-way ANOVA. Error bars = SEM; $N = 3$ biological repeats. * = $P \le 0.05$, ns = not significant $p > 0.05$. Statistical values for NTC vs siSAE $P = 0.0060$, siSAE vs siSAE+SAE2-WT $P = 0.0064$, NTC vs siSAE+SAE2-WT $P > 0.9999$, siSAE+SAE2-WT vs siSAE+SAE2-C173G $P = 0.0174$, siSAE+SAE2-WT vs siSAE+SAE2-K164Q $P = 0.9665$, siSAE+SAE2-WT vs siSAE+SAE2-K164R $P = 0.4874$. (D) Automated analysis of γH2AX foci numbers, obtained through high-content microscopy, in U2OS cells treated with indicated siRNAs (siNTC- black, siSAE1:SAE2- red) with or without the complementation of inducible siRNA-resistant SAE2 variants (SAE2 WT- dark blue, SAE2 CG- orange, SAE2 K164Q- grey, SAE2 K164- light blue). siUBC9 (purple) and siPIAS1 (green) are used for comparison. Results displayed for data isolated from S phase (top), G1 (middle) and G2 (bottom) cell populations. Plotted data is derived from the mean number of foci per condition from 3 independent biological repeats, error bars = SEM. Statistical significance was calculated using two-way ANOVA using Dunnett's multiple comparisons test. Timepoints where there is a significant difference from the non-target control siRNA condition are marked with * = $P < 0.05$, ** = $P < 0.01$, *** = $P < 0.001$. Purple and green * show that only siUBC9 and siPIAS1 conditions significantly deviate from siNTC at points in the time course. siNTC vs siSAE1:SAE2 (S phase 1 h $P = 0.8997$, 6 h $P = 0.7437$; G1 1 h $P = 0.7623$, 6 h $P = 0.9997$; G2 1 h $P = 0.9993$, 6 h $P = 0.9988$), siNTC vs SAE2-WT (S phase 1 h $P = 0.9792$, 6 h $P > 0.9999$; G1 1 h $P = 0.9997$, 6 h $P = 0.7655$; G2 1 h $P = 0.9971$, 6 h $P = 0.7660$), siNTC vs SAE2-CG (S phase 1 h $P = 0.9907$, 6 h $P > 0.9999$; G1 1 h $P = 0.9965$, 6 h $P = 0.9989$; G2 1 h $P = 0.9957$, 6 h $P = 0.9494$), siNTC vs SAE2-KQ (S phase 1 h $P = 0.9686$, 6 h $P = 0.9993$; G1 1 h $P = 0.9978$, 6 h $P = 0.9932$; G2 1 h $P > 0.9999$, 6 h $P > 0.9999$), siNTC vs SAE2-KR (S phase 1 h $P = 0.8283$, 6 h $P > 0.9211$; G1 1 h $P = 0.9430$, 6 h $P = 0.9799$; G2 1 h $P = 0.9999$, 6 h $P = 0.9967$), siNTC vs siUBC9 (S phase 1 h $P = 0.0076$, 6 h $P = 0.0086$; G1 1 h $P = 0.0277$, 6 h $P = 0.9833$; G2 1 h $P = 0.0073$, 6 h $P = 0.5598$), siNTC vs siPIAS1 (S phase 1 h $P = 0.0016$, 6 h $P < 0.0001$; G1 1 h $P = 0.0002$, 6 h $P = 0.0020$; G2 1 h $P = 0.0002$, 6 h $P = 0.0051$). (E) The measure of DNA repair from U2OS cells bearing integrated DNA repair reporters in cells treated with siNTC or siSAE1:siSAE2 and transfected with the enzyme, I-SCE-1. Illustration of the integrated DNA repair substrates for homologous recombination and non-homologous end-joining (Top). The graph (Bottom) displays the percentage of GFP-positive cells normalised to RFP-transfection efficiency. %-repair of siSAE1:SAE2 is given relative to siNTC. Data from 2 independent biological repeats. (F) Immunoprecipitation of endogenous mitotic SUMO conjugates in U2OS cells treated with ML792 and expressing Flag-SAE2 constructs resistant to the inhibitor. ML792 resistance is denoted by (r). The presence of Flag-SAE2r is represented by (WTr), Flag-SAE2r-K164Q by KQr, and Flag-SAE2r-K164R by KRr. The diagram, top, illustrates the timing of inhibitor and induction agent addition. To better detect free SUMO, Y299 (SUMO1) and 8A2 (SUMO2/3) antibodies were employed (Garvin et al, 2022). Performed once. (G) The percentage of micronuclei positive for γH2AX in asynchronous siRNA-resistant SAE2 variant-expressing U2OS cells. Error bars SEM. Data from 4 independent biological repeats. $N > 50$ micronuclei per condition per biological repeat. Significance was tested using one-way ANOVA no significant differences between conditions were identified. (H) Western blot analysis of U2OS treated with nocodazole ±5 μM ML792 for 16 h. Mitotic cells were harvested by mitotic shake-off and lysed in loading buffer, and western probed for NuMA and SAE2. Performed twice. (I) Representative images validating the specificity of the NuMA antibody. NuMA colocalises to -β tubulin metaphase cells adjacent to the DAPI stain. NuMA signal significantly diminished after 72-h 10 nM siNuMA treatment. Cells chosen from >50 similar cells, performed once. White bar indicates 5 micrometers.

                                                                                                                                                                     

**A**

```
CLUSTAL O(1.2.4) multiple sequence alignment

sp|P0CG47|UBB_HUMAN      --------------------MQIFVKTLTGKTITLEVEPSDTIENVKAKIQDKEGIPPD 39
sp|Q15843|NEDD8_HUMAN    --------------------MLIKVKTLTGKEIEIDIEPTDKVERIKERVEEKEGIPPQ 39
sp|P63165|SUMO1_HUMAN    MSDQEAKPSTEDLGDKKEGEYIKLKVIGQDSSEIHFKVKMTTHLKKLKESYCQRQGVPMN 60
sp|P61956|SUMO2_HUMAN    MADEKPK----EGVKTENNDHINLKVAGQDGSVVQFKIKRHTPLSKLMKAYCERQGLSMR 56
sp|P55854|SUMO3_HUMAN    MSEEKPK----EGVKTE-NDHINLKVAGQDGSVVQFKIKRHTPLSKLMKAYCERQGLSMR 55
                                            : : *    .. : :.::   :..:       :::*:

sp|P0CG47|UBB_HUMAN      QQRLIFAGKQLEDGRTLSDYNIQKESTLHLVLRLRGG    76
sp|Q15843|NEDD8_HUMAN    QQRLIYSGKQMNDEKTAADYKILGGSVLHLVLALRGG    76
sp|P63165|SUMO1_HUMAN    SLRFLFEGQRIADNHTPKELGMEEEDVIEVYQEQTGG    97
sp|P61956|SUMO2_HUMAN    QIRFRFDGQPINETDTPAQLEMEDEDTIDVFQQQTGG    93
sp|P55854|SUMO3_HUMAN    QIRFRFDGQPINETDTPAQLEMEDEDTIDVFQQQTGG    92
                          . *: : *: : :  *  : :   ..:.:    **
```

**B**

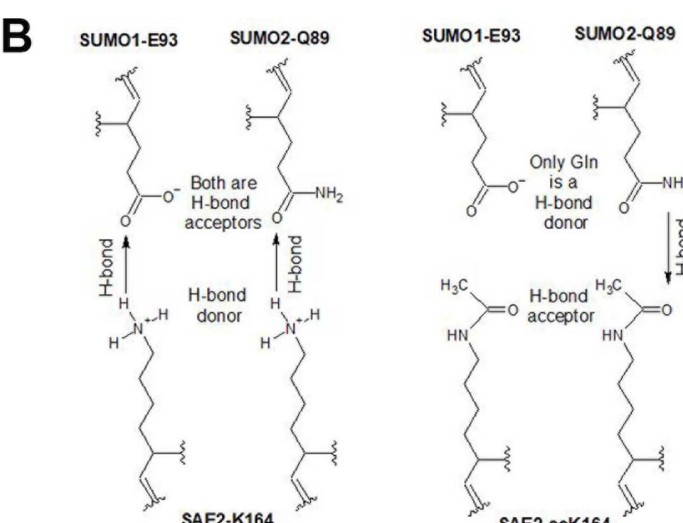

**Figure EV5. Discrimination between SUMO1 and SUMO2 by SAE2-K164.**

(A) Amino acid sequence alignment for Ubiquitin (UBB), Nedd8, SUMO1, SUMO2, and SUMO3 up to the C-terminal di-Gly motif representative of mature activation/conjugation-competent Ubls. The black box indicates Ubiquitin-R72, which is divergent in Nedd8, SUMO1, and SUMO2/3 and required for Ubl E1 discrimination of different Ubl modifiers (Walden et al, 2003). Generated using Uniprot and Clustal Omega. (B). Illustration of one proposed mechanism of SAE1:SAE2-acK164 bias for SUMO2-Q89 over SUMO1-E93 through hydrogen bonding patterns in the SUMO E1 'closed' conformation. Unacetylated SAE2-K164 acts as a hydrogen bond donor to both SUMO1-E93 or SUMO2-Q89, SAE2-acK164 acquires hydrogen bond acceptor capacity to which SUMO2-Q89 donate a hydrogen bond, while SUMO1-E93 cannot. This hypothetical hydrogen bonding arrangement explains why SAE1:SAE2-acK164 bears a bias towards SUMO2 activation. A second, additive model, is that of electrostatic interaction between SAE2-K164 and SUMO1-E93, which is absent for K164-SAE2:Q89-SUMO2 and lost for acK164-SAE2:E93-SUMO1/Q89-SUMO2 context.

