## [Peer Review File · The EMBO Journal]

HDAC6-Dependent Deacetylation of SAE2 enhances SUMO1 Conjugation for Mitotic Integrity.

Alexander Lanz, Alexandra Walker, Mohammed Jamshad, Alexander Garvin, Matthew Stewart, Peter Wotherspoon, Benjamin Cooper, Matt MacKintosh, Oliver Crutchley, Timothy Knowles, and Joanna Morris

Corresponding author(s): Joanna Morris (j.morris.3@bham.ac.uk)

Review Timeline:

Transfer from Review Commons:	10th Feb 25
Editorial Decision:	20th Feb 25
Revision Received:	7th Jun 25
Editorial Decision:	14th Jul 25
Revision Received:	24th Jul 25
Accepted:	25th Jul 25

Review
COMMONS

Editor: Hartmut Vodermaier

Transaction Report: This manuscript was transferred to The EMBO JOURNAL following peer review at Review Commons.

Review #1

1. Evidence, reproducibility and clarity:

Evidence, reproducibility and clarity (Required)

Summary:

In their manuscript, Walker et al. investigate the physiological role of deacetylation of the SAE2 subunit of the SUMO E1 enzyme. They find that SAE1:SAE2-acK164 is deacetylated in an HDAC6-dependend manner and use a series of biochemical assays to show that deacetylation of the SAE2 subunit shifts the bias of the SUMO E1 towards SUMO1 conjugation in vitro, proposing a mechanism that is similar to the one that the NEDD8 E1 employs to discriminate between NEDD8 and ubiquitin.

The authors continue to examine the role of different SAE2 variants in different cellular stresses and show that the acetyl-mimicking SAE2K164Q variant displays reduced levels of high molecular weight SUMO1 conjugates in mitotic cells. This variant cannot support proper mitotic spindle formation leading to the appearance of multipolar spindles and centromere-containing micronuclei. Finally, they go on to identify the mechanism underlying these phenotypes and examine NuMA SUMOylation. They test SUMOylation-refractive NuMA variants as well as an already published SUMO1-NuMA fusion that mimics the SUMOylated protein form. They propose a model in which deacetylation of SAE2 changes the bias of the SUMO E1 to increase SUMO1-NuMA conjugation during mitosis, promoting bipolar spindle formation.

Major point:

As the authors state, SUMO1 conjugates decrease during mitosis and this is somewhat at odds with the proposed model regarding NuMA. The authors can detect a SUMOylated NuMA conjugate (fig. 4a). To test whether the proposed model is correct, the authors could check:

- a. Whether this form is indeed SUMO1-NuMA
- b. Whether it decreases upon expression of the SAE2K164Q variant.

Minor points:

1. Fig. 2c: Why does C173G form a thioester with SUMO2 up to 40% of the WT?
2. Please clarify the use of Dox addition in the text and legend earlier (is found currently in Supp. Fig 4).
3. Fig. 4f: what is the difference between the first (invisible NUMA) bipolar and the second, NuMA visible bipolar spindle?
4. ML972- should read ML792 on pg 8.

2. Significance:

Significance (Required)

General assessment:

This is a thorough study with complex but well controlled experiments and contains a large amount of valuable information. A point could be further clarified in order to provide further support the proposed model.

Advance:

The document brings understanding on the regulation of the SUMO conjugation system a step forward and links it to a physiological context.

Audience:

basic science: the Ubiquitin family field and also the mitosis-cytoskeleton field. Applied science concerning the use of SUMO inhibitors in cancer.

*Expertise: SUMO regulation of the cytoskeleton during mitosis (yeast system)

3. How much time do you estimate the authors will need to complete the suggested revisions:

Estimated time to Complete Revisions (Required)

(Decision Recommendation)

Between 1 and 3 months

4. Review Commons values the work of reviewers and encourages them to get credit for their work. Select 'Yes' below to register your reviewing activity at Web of Science Reviewer Recognition Service (formerly Publons); note that the content of your review will not be visible on Web of Science.

Yes

Review #2

1. Evidence, reproducibility and clarity:

Evidence, reproducibility and clarity (Required)

****Summary:****

Walker et al characterized lysine 164 acetylation of the catalytic SUMO activating enzyme subunit SAE2 and observed that this modification causes a bias towards SUMO2/3 over SUMO1 involving their C-terminal tails. While several enzymes appear to mediate SAE2 acetylation, HDAC6 is responsible for deacetylating SAE2 in mitosis, thereby promoting mitotic SUMO1 modification. The nuclear mitotic apparatus, NuMA, was identified as a putative mitotic SUMO1 substrate upon SAE2 deacetylation. Replacement of endogenous SAE2 with an acetylation mimetic SAE2-K164Q mutant restricts SUMO1 conjugation of NuMA resulting in multipolar spindle formation that can be rescued either by overexpression of SUMO1 or by SUMO1-NuMA fusion.

****Major comments:****

- Figures 2 C/Supplementary Figure 3c: The enzyme concentrations used in these reactions are much too high. To discriminate between thioester- and isopeptide-linked SUMO, the same samples should be analyzed in the absence (detection of thioester and isopeptide linkages) and presence of high concentrations of DTT (detection of isopeptide-linked SUMO only). The presented assay is problematic as it shows dimeric SUMO and RanGAP1:SUMO bands in the absence of ATP and no UBC9 but SAE2 thioester/isopeptide formation in the absence of RanGAP1 (preferentially UBC9 should form a thioester/isopeptide bond in this condition as higher molarities of UBC9 over E1 are used). Dimeric SUMO should not be detected unless disulfide bridges are formed between cysteines - this happens when DTT is not present in the reaction - under such conditions, SAE2 and UBC9 can also form disulfide bridges via their catalytic cysteines, impairing their enzymatic activity. In order to interpret the results correctly, it is important to add low concentrations of DTT (~0.1 mM) even in thioester reactions and to distinguish between thioester and isopeptide linkages.

- Figure 2F/ Supplementary Figure 3d: Again, the enzyme concentrations are much too high

and need to be reduced to a concentration where mainly RanGAP1 monosumoylation with SUMO1 is detected. As RanGAP1 is the most efficient SUMO substrate known, the enzyme concentrations and reaction time can be greatly reduced to limit the auto-modification of the enzymes and SUMO chain formation. Due to the efficient chain-forming activity of SUMO2, this is more difficult with SUMO2, but can be reduced by limiting the concentration of UBC9 in particular or by using a SUMO2 KallR mutant. In the reaction shown, the authors used only twice the molarity of SUMO compared to the substrate, too low taking into account SUMO2 chain formation, enzyme and substrate modification (The reaction should be limited by enzyme activity not by SUMO2). How can the authors be sure that the band they report as RanGAP1 high MW SUMO2 is indeed RanGAP1 modified and not SAE2 (in comparison to Suppl Figure 3b)?

- Figure 3 nicely shows that ML792-resistant SAE2 variants conjugate SUMO2 equally well, whereas SAE2 K163R is reduced and SAE2 K163Q appears to be abolished in SUMO1 conjugation. However, only high molecular weight SUMO conjugates are shown. What are the levels of free SUMO after overexpression of SAE2 variants and the indicated treatments? According to the work of Zhang et al from the Matunis lab (cited as reference 39 in the proposed study), SUMO conjugation is greatly reduced in nocodazole-arrested cells, but is restored after release in G1. Furthermore, SUMO1 and SUMO2 localize to different subcellular regions during mitosis. Have the authors tested whether SAE2 variants differ in their intracellular localization or alter the subcellular localization of SUMO1 and SUMO2 in interphase and mitotic cells? Can the authors comment on the proportion of SAE2 that is acetylated?

- Figure 4: The authors show a ML792 sensitive high molecular weight smear of NUMA in nocodazole treated cells. It would be very convincing if the authors could demonstrate whether endogenous NUMA is conjugated to SUMO1 or SUMO2 in mitosis by SUMO IPs and whether they can detect a change upon expression of SAE2 variants as in Figure 3a. By replicating this experiment, it would be important to demonstrate the presence of both free and conjugated SUMO paralogs in the input and paralog specific sumoylation in general (smear) and of NUMA in the IP.

****Minor comments:****

- Supplementary Figure 2: Please indicate the size of the marker bands, the fraction numbers and which fractions were pooled for further analysis. Is there any explanation why SAE1:SAE2K164R eluates in two peaks, suggesting two complexes? How different are they in size?

2. Significance:

Significance (Required)

The finding that E1 acetylation regulates SUMO paralog specificity is very exciting, particularly because of its link to key regulatory mitotic functions. Overall, the findings are intriguing and supported in part by various biological and biochemical methods. However, some concerns remain unsatisfactorily addressed, as outlined above.

The findings provide a novel basic concept of how E1 enzyme regulation contributes to the specification of modifier selectivity, demonstrates cross-talk with other PTMs and reveals a biological function. Therefore, the study is of interest to a broad audience.

3. How much time do you estimate the authors will need to complete the suggested revisions:

Estimated time to Complete Revisions (Required)

(Decision Recommendation)

Between 1 and 3 months

Yes

Review #3

1. Evidence, reproducibility and clarity:

Evidence, reproducibility and clarity (Required)

****Summary:****

In this manuscript, the authors report on an interesting regulatory mechanism that influences the balance between conjugation of the different SUMO isoforms, SUMO1 versus SUMO2/3. The authors describe that acetylation of a specific residue, K164, in the SUMO activating enzyme (E1) subunit, SAE2, biases the E1's preference towards SUMO2/3. Specifically, they use an acetylation-mimicking K164Q mutation to show that the

acetylation state of SAE2 likely affects the affinity of the E1 to SUMO and the rate of thioester formation. With an antibody, they demonstrate the acetylation of SAE2 in cells. Mechanistically, they locate the cause of the isoform bias to a residue in the C-terminus of SUMO in proximity to K164 or SAE2, where SUMO1 carries glutamate, while SUMO2/3 has glutamine. Switching these residues between the SUMO isoforms reverses the isoform preference of the E1. Phenotypically, the SAE2 K164Q mutant induces mitotic problems that the authors attribute to the SUMOylation of the NuMA complex. They assign the deacetylation of SAE1 to HDAC6 and report that deacetylation occurs during mitosis. These results are consistent with a model that SUMO1 modification of the NuMA complex in mitosis is important for mitotic fidelity and that the cell cycle-dependent changes in the acetylation status of SAE2 promote this. Accordingly, fusion of SUMO1 to a NuMA subunit partially overcomes the problems induced by the K164Q mutant or the inhibition of HDAC6.

****Major comments:****

The experiments are largely performed in a well-controlled manner, and overall, the study is very convincing. I would like to suggest a few experiments that would strengthen the authors' conclusions, and there are a few minor issues with some of the figures.

1. It would be helpful if the authors could more clearly separate the two steps catalyzed by the E1. This would be needed to determine whether the accumulation of the SUMO1-AMP intermediate by the K164Q mutant is due to a faster rate of formation or a reduced rate of conversion to the thioester. They could test the AMP formation step in isolation in a straightforward manner by using the double mutant K164Q C173G and measuring a time course of SUMO1-AMP versus SUMO2-AMP build-up. Alternatively, they could try to isolate the second step by adding SUMO1-AMP versus SUMO2-AMP to the E1 de novo - although isolation of the intermediates may be more involved.
2. The reason for the isoform selectivity in the context of NuMA SUMOylation remains unresolved. The study would be significantly strengthened if the authors could address the question of whether the mitotic defects come from a lack of NuMA SUMOylation or the wrong type of SUMOylation. In other words, does it matter which isoform of SUMO is attached to NuMA? This could be addressed by also creating a SUMO2 fusion construct and testing whether that suppresses some of the phenotypes observed with the K164Q mutant and upon HDAC6 inhibition.
3. The authors should include a more detailed discussion of the importance of the absolute and relative concentrations of free SUMO1 versus SUMO2/3 as a possible mechanism to impose isoform bias. Specifically, they should consider the different KM values of the E1 for

the isoforms. The literature says that the E1 has a lower K_M (higher affinity) for SUMO1 than SUMO2/3 but also a lower k_{cat} (considering both steps of its reaction together), resulting in an approximately equal k_{cat}/K_M . This would mean that at low overall SUMO concentrations, SUMO1 would have an advantage, whereas with rising SUMO concentrations SUMO2/3 would be favoured (which might be particularly important during stress conditions). What part of the curve does the cellular environment reflect?

4. It would be helpful to show a time course of endogenous SAE2 acetylation over the cell cycle, using synchronized cultures. All the experiments showing acetylation are done with transfected FLAG-tagged constructs - are they overexpressed? Is it not possible to work with endogenous SAE2?

****Minor comments:****

- The title is not immediately understandable. "SUMO protein bias for mitotic stability" sounds a bit awkward. It would be clearer to be more explicit about isoforms.
- On page 3, the authors could introduce a justification of why they tested IR treatment.
- The authors repeatedly use the word "codon" when they describe a site in the protein. Codon refers to mRNA, so the word "residue" would be more appropriate when talking about a protein.
- Page 8: "confirmation" should be "conformation".
- Page 8: "While we find a little role for..." - delete "a"
- Fig 2a: The figure would be easier to understand if the same colour scheme was used for S1 versus S2 to aid the comparison.
- Fig 2b: I don't understand the units of this graph. Why does normalization result in a value of zero, not 1? On this scale, what would a value of 1 signify? How can a value become negative? I would have expected values relative to the WT, with the WT being set to 1 or to 100%. The authors should also show the raw data for this plot.
- Fig 2c: Please also show representative raw data.
- Fig 2d,f: Again, the legend should explain what the plots were normalized to.
- Fig 3g: Could the authors comment on the detrimental effects of both SUMO1 and SUMO2 in the WT background?
- Fig 3h: typo ("Trasfect")
- Fig 4b: The images have very poor contrast. In addition, the merged image would be clearer if two different colours were used.
- Fig 4f: The DAPI signal is hardly visible - better contrast would help.
- Fig S2: It would be appropriate to indicate which fractions were actually collected or combined during the purification.
- Fig S5b: The authors argue with the hydrogen bonding capacities of the different pairings.

However, acetylation at K164 should not necessarily prevent a hydrogen bond to SUMO1-E93, considering that the "NH" group is likely still at a comparable distance to the carboxylate of E93 and could in principle undergo H-bonding unless prevented by the steric bulk introduced by the acetyl group. On the other hand, the K164-E93 interaction is the only electrostatic interaction among the 4 possible combinations. While a contribution is not easy to prove experimentally, I think the possibility of charge-charge interactions having an impact should be considered in the discussion.

2. Significance:

Significance (Required)

The results presented here are interesting and novel. Importantly, the authors provide a molecular model for a new mechanism of how the SUMO system achieves isoform specificity, which is a still very poorly understood phenomenon. The manuscript makes a significant advance by contributing an important new aspect of how the SUMO conjugation machinery chooses between isoforms. The manuscript is strong by providing very good evidence for its conclusions. One limitation is the strong reliance on the use of an acetyl-mimicking mutant; this limitation could be overcome by placing a bit more emphasis on detecting endogenous SAE2 acetylation.

Audience: The study should be relevant to a broad audience, given the impact of the SUMO system on cellular regulation; after all, the study addresses a very fundamental problem in the field. In addition, it should be of interest to researchers studying regulation of mitosis.

3. How much time do you estimate the authors will need to complete the suggested revisions:

Estimated time to Complete Revisions (Required)

(Decision Recommendation)

Between 1 and 3 months

Yes

Revision Plan

Manuscript number:

RC-2024-02794

Corresponding author(s): Jo Morris

[The “revision plan” should delineate the revisions that authors intend to carry out in response to the points raised by the referees. It also provides the authors with the opportunity to explain their view of the paper and of the referee reports.]

The document is important for the editors of affiliate journals when they make a first decision on the transferred manuscript. It will also be useful to readers of the reprint and help them to obtain a balanced view of the paper.

*If you wish to submit a full revision, please use our "Full Revision" template. **It is important to use the appropriate template to clearly inform the editors of your intentions.**]*

1. General Statements [optional]

We feel the reviewers understood the paper well and made many reasonable points for improvement.

In response to Reviewer three’s concern about the reliance on SAE2 over-expression, in the ‘Significance’ section “*One limitation is the strong reliance on the use of an actyl-mimicking mutant*”. We were minded not to rely on the mutant. Hence, the paper contains considerable data on the HDAC6 deacetylase, responsible for SAE2 deacetylation. We show that HDAC6 inhibition phenocopies SAE2-K164Q expression and, moreover, that the approaches which rescue the mitotic defects of SAE2-K164Q expression cells also rescue the defects of HDAC6 inhibited cells. These observations, we believe, overcome the concern.

2. Description of the planned revisions

Insert here a point-by-point reply that explains what revisions, additional experimentations and analyses are planned to address the points raised by the referees.

Revisions.

R1: As the authors state, SUMO1 conjugates decrease during mitosis and this is somewhat at odds with the proposed model regarding NuMA. The authors can detect a SUMOylated NuMA conjugate (fig. 4a). To test whether the proposed model is correct, the authors could check:

- a. Whether this form is indeed SUMO1-NuMA
- b. Whether it decreases upon expression of the SAE2K164Q variant.

Revision Plan

R2: Figure 4: The authors show a ML792 sensitive high molecular weight smear of NUMA in nocodazole treated cells. It would be very convincing if the authors could demonstrate whether endogenous NUMA is conjugated to SUMO1 or SUMO2 in mitosis by SUMO IPs and whether they can detect a change upon expression of SAE2 variants as in Figure 3a. By replicating this experiment, it would be important to demonstrate the presence of both free and conjugated SUMO paralogs in the input and paralog specific sumoylation in general (smear) and of NUMA in the IP.

Response: These are important points. We intend to perform the suggested experiments to address which isoform NuMa is modified by, and what the impact of the variant is.

R2: Figures 2 C/Supplementary Figure 3c: The enzyme concentrations used in these reactions are much too high. To discriminate between thioester- and isopeptide-linked SUMO, the same samples should be analyzed in the absence (detection of thioester and isopeptide linkages) and presence of high concentrations of DTT (detection of isopeptide-linked SUMO only). The presented assay is problematic as it shows dimeric SUMO and RanGAP1:SUMO bands in the absence of ATP and no UBC9 but SAE2 thioester/isopeptide formation in the absence of RanGAP1 (preferentially UBC9 should form a thioester/isopeptide bond in this condition as higher molarities of UBC9 over E1 are used). Dimeric SUMO should not be detected unless disulfide bridges are formed between cysteines - this happens when DTT is not present in the reaction - under such conditions, SAE2 and UBC9 can also form disulfide bridges via their catalytic cysteines, impairing their enzymatic activity. In order to interpret the results correctly, it is important to add low concentrations of DTT (~0.1 mM) even in thioester reactions and to distinguish between thioester and isopeptide linkages.

R2: Figure 2F/ Supplementary Figure 3d: Again, the enzyme concentrations are much too high and need to be reduced to a concentration where mainly RanGAP1 monosumoylation with SUMO1 is detected. As RanGAP1 is the most efficient SUMO substrate known, the enzyme concentrations and reaction time can be greatly reduced to limit the auto-modification of the enzymes and SUMO chain formation. Due to the efficient chain-forming activity of SUMO2, this is more difficult with SUMO2, but can be reduced by limiting the concentration of UBC9 in particular or by using a SUMO2 KallR mutant. In the reaction shown, the authors used only twice the molarity of SUMO compared to the substrate, too low taking into account SUMO2 chain formation, enzyme and substrate modification (The reaction should be limited by enzyme activity not by SUMO2). How can the authors be sure that the band they report as RanGAP1 high MW SUMO2 is indeed RanGAP1 modified and not SAE2 (in comparison to Suppl Figure 3b)?

Response: We intend to repeat these assays, as suggested by the reviewer, reducing the enzyme concentrations and using low-concentration DTT. With the relevant controls and blots to show the identity of the RanGAP-SUMO2 product. Further, we will show control experiments with and without DTT that demonstrate the sensitivity of the SAE2-SUMO band to the reducing agent.

Revision Plan

R2: Figure 3 nicely shows that ML792-resistant SAE2 variants conjugate SUMO2 equally well, whereas SAE2 K163R is reduced and SAE2 K163Q appears to be abolished in SUMO1 conjugation. However, only high molecular weight SUMO conjugates are shown. What are the levels of free SUMO after overexpression of SAE2 variants and the indicated treatments?

Response: We will attempt to show free SUMO levels in mitotic cells.

R2: According to the work of Zhang et al from the Matunis lab (cited as reference 39 in the proposed study), SUMO conjugation is greatly reduced in nocodazole-arrested cells, but is restored after release in G1. Furthermore, SUMO1 and SUMO2 localize to different subcellular regions during mitosis. Have the authors tested whether SAE2 variants differ in their intracellular localization or alter the subcellular localization of SUMO1 and SUMO2 in interphase and mitotic cells?

Response: We will examine the localisation of the SAE2 variants (see section below for the SUMO proteins).

R3: It would be helpful if the authors could more clearly separate the two steps catalyzed by the E1. This would be needed to determine whether the accumulation of the SUMO1-AMP intermediate by the K164Q mutant is due to a faster rate of formation or a reduced rate of conversion to the thioester. They could test the AMP formation step in isolation in a straightforward manner by using the double mutant K164Q C173G and measuring a time course of SUMO1-AMP versus SUMO2-AMP build-up. Alternatively, they could try to isolate the second step by adding SUMO1-AMP versus SUMO2-AMP to the E1 de novo - although isolation of the intermediates may be more involved.

Response: We intend to perform the first approach suggested, making and examining the double mutant's activity as suggested.

R3: The reason for the isoform selectivity in the context of NuMA SUMOylation remains unresolved. The study would be significantly strengthened if the authors could address the question of whether the mitotic defects come from a lack of NuMA SUMOylation or the wrong type of SUMOylation. In other words, does it matter which isoform of SUMO is attached to NuMA? This could be addressed by also creating a SUMO2 fusion construct and testing whether that suppresses some of the phenotypes observed with the K164Q mutant and upon HDAC6 inhibition.

Response. This is an excellent suggestion. We intend to make the constructs suggested and perform this experiment for our revision.

Revision Plan

R3: It would be helpful to show a time course of endogenous SAE2 acetylation over the cell cycle, using synchronized cultures.

Response: We will attempt to gain a view of SAE2 acetylation over the cell cycle, which requires the precipitation of endogenous SAE following synchronisation.

R3: Fig 2a: The figure would be easier to understand if the same colour scheme was used for S1 versus S2 to aid the comparison.

Response: We will change this.

R3: The title is not immediately understandable. "SUMO protein bias for mitotic stability" sounds a bit awkward. It would be clearer to be more explicit about isoforms.

Response: We have considered: "HDAC6-Dependent Deacetylation of SAE2 enhances SUMO1 Conjugation for Mitotic Integrity", we have not changed it on the current manuscript so as not to confuse the reader - we will change it at the journal level.

R3: Fig 2b: I don't understand the units of this graph. Why does normalization result in a value of zero, not 1? On this scale, what would a value of 1 signify? How can a value become negative? I would have expected values relative to the WT, with the WT being set to 1 or to 100%. The authors should also show the raw data for this plot.

Response: The data will be normalised to the WT condition (1 instead of 0), and raw data shown.

R3: Fig 2c: Please also show representative raw data.

Response: Representative images will be shown.

R3: Fig 2d,f: Again, the legend should explain what the plots were normalized to.

Response: Inserted in the legend for Fig. 2d&f: "The RanGAP1-SUMO1 products are normalised to the WT SAE1:SAE2:SUMO1-only condition (top) and the RanGAP1-SUMO2 products are normalised to the WT SAE1:SAE2:SUMO2-only condition (bottom)."

R3 Fig S5b: The authors argue with the hydrogen bonding capacities of the different pairings. However, acetylation at K164 should not necessarily prevent a hydrogen bond to SUMO1-E93, considering that the "NH" group is likely still at a comparable distance to the carboxylate of E93 and could in principle undergo H-bonding unless prevented by the steric bulk introduced by the acetyl group. On the other hand, the K164-E93 interaction is the only electrostatic interaction

Revision Plan

among the 4 possible combinations. While a contribution is not easy to prove experimentally, I think the possibility of charge-charge interactions having an impact should be considered in the discussion.

Response: Agreed. The figure will be redrawn, and the possibility will be discussed.

R1 Fig. 2c: Why does C173G form a thioester with SUMO2 up to 40% of the WT?

Response: We believe this arose in measuring background density in the blots in error. We will re-assess the method used.

R3: Fig 4b: The images have very poor contrast. In addition, the merged image would be clearer if two different colours were used.

Response: We will change one of the colours.

3. Description of the revisions that have already been incorporated in the transferred manuscript

Please insert a point-by-point reply describing the revisions that were already carried out and included in the transferred manuscript. If no revisions have been carried out yet, please leave this section empty.

R1.2. Please clarify the use of Dox addition in the text and legend earlier (is found currently in Supp. Fig 4).

Response: Inserted before first result using doxycycline: "Furthermore, we generated U2OS with a doxycycline-inducible (wild-type) WT FLAG-SAE2 or a FLAG-SAE2-K164R mutant."

R1.3. Fig. 4f: what is the difference between the first (invisible NUMA) bipolar and the second, NuMA visible bipolar spindle?

Response: Fig. 4f now annotated with 'Untransfected' and 'GFP-NuMA transfected'.

R1.4. ML972- should read ML792 on pg 8.

Response: Corrected.

R3: All the experiments showing acetylation are done with transfected FLAG-tagged constructs - are they overexpressed?

Response: Supplemental Figure 4a illustrates that with the exception of the C173G mutant, the remainder WT, and K164-mutants are all expressed at near WT-levels and not over-expressed. The C-G-mutant is highly expressed.

Revision Plan

R3: On page 3, the authors could introduce a justification of why they tested IR treatment.

Response: now justified.

R3: The authors repeatedly use the word "codon" when they describe a site in the protein. Codon refers to mRNA, so the word "residue" would be more appropriate when talking about a protein.

Response: Agreed. Done.

R3: Page 8: "confirmation" should be "conformation".

Response: Done.

R3: Page 8: "While we find a little role for..." – delete "a"

Response: Done.

R2: Supplementary Figure 2: Please indicate the size of the marker bands, the fraction numbers and which fractions were pooled for further analysis. Is there any explanation why SAE1:SAE2K164R eluates in two peaks, suggesting two complexes? How different are they in size?

Response: Ladder markers added to each gel image. Fraction numbers added. Black box indicates fractions pooled. Figure updated with relevant recombinant protein preps generated for updated in vitro experiments. The additional SAE1:SAE2-K164R peak which appeared in the previous manuscript Supp. Fig. 2a eluted in the void volume and so we think it comprised aggregated SAE1:SAE2 protein, more recent preparations do not show it.

R3: The authors should include a more detailed discussion of the importance of the absolute and relative concentrations of free SUMO1 versus SUMO2/3 as a possible mechanism to impose isoform bias. Specifically, they should consider the different K_M values of the E1 for the isoforms. The literature says that the E1 has a lower K_M (higher affinity) for SUMO1 than SUMO2/3 but also a lower k_{cat} (considering both steps of its reaction together), resulting in an approximately equal K_{cat}/K_M . This would mean that at low overall SUMO concentrations, SUMO1 would have an advantage, whereas with rising SUMO concentrations SUMO2/3 would be favoured (which might be particularly important during stress conditions). What part of the curve does the cellular environment reflect?

Response: Yes, good point. Now included:

R3: Fig 3g: Could the authors comment on the detrimental effects of both SUMO1 and SUMO2 in the WT background?

Response: Comment included.

R3: Fig 3h: typo ("Trasfect")

Revision Plan

Response: Done.

R3: Fig 4f: The DAPI signal is hardly visible - better contrast would help.

Response: Improved.

R3: Fig S2: It would be appropriate to indicate which fractions were actually collected or combined during the purification.

Response: Ladder markers added to each gel image. Fraction numbers added. Black box indicates the fractions pooled.

4. Description of analyses that authors prefer not to carry out

Please include a point-by-point response explaining why some of the requested data or additional analyses might not be necessary or cannot be provided within the scope of a revision. This can be due to time or resource limitations or in case of disagreement about the necessity of such additional data given the scope of the study. Please leave empty if not applicable.

R2: According to the work of Zhang et al from the Matunis lab (cited as reference 39 in the proposed study), SUMO conjugation is greatly reduced in nocodazole-arrested cells, but is restored after release in G1. Furthermore, SUMO1 and SUMO2 localize to different subcellular regions during mitosis. Have the authors tested whether SAE2 variants differ in their intracellular localization or alter the subcellular localization of SUMO1 and SUMO2 in interphase and mitotic cells?

Response: We have investigated SUMO isoform location. However, in our hands, using a range of SUMO antibodies, we do not see the previously reported localisations in mitotic wild-type cells, and thus, we are not able to assess the impact of the SAE variants. As our phenotypes are restricted to mitosis, we do not consider it worthwhile to look at interphase.

Prof. Joanna R Morris
University of Birmingham
School of Cancer Sciences
College of Medical and Dental Schools
Birmingham, West Midlands B15 2TT
United Kingdom

20th Feb 2025

Re: EMBOJ-2025-120456-T

The SUMO activating enzyme subunit, SAE2, contributes SUMO protein bias for mitotic fidelity.

Dear Jo,

Thank you for transferring your manuscript from Review Commons to The EMBO Journal. I have now had the chance to read your study, as well as to go through the referee reports and your responses to them. Given the interest of the subject and your findings, which has also been acknowledged by the reviewers, we would be happy to consider an adequately revised version further for EMBO Journal publication. I realize that the referees raised a number of well-taken experimental concern, and appreciate that you plan to comprehensively address most of them through follow-up experiments; with the final judgement of the referees obviously depending to some degree on the outcome of these revision experiments, and whether they are able to validate the main conclusions of the study. In this light, I would like to formally invite you to revise the manuscript along the lines suggested in your response letter.

When preparing a revised manuscript, please try to adhere to the guidelines listed below and in our Guide to Authors as closely as possible, as this should greatly facilitate our assessment at the time of resubmission - in particular regarding the completion of our author checklist and a dedicated reagents and tools table (both linked below), the inclusion of editable text files and separate, individual figures, the formatting of the references, and the conversion of "supplemental" material into Expanded View and/or Appendix content. Please also note that it is our policy to allow only a single round of (major) revision, making it important to comprehensively answer all criticisms at this point - if this should require more time than our standard three-months deadline, I would be happy to discuss an extension of the revision time, during which our 'scoping protection' (meaning that competing work appearing elsewhere in the meantime will not affect our considerations of your study) would of course remain valid. Please do not hesitate to contact me should you have any further questions at this stage.

Thank you again for the opportunity to consider this study for The EMBO Journal. I look forward to receiving your revision.

With kind regards,

Hartmut

9) To facilitate reproducibility and cross-laboratory adoption of methodologies, please structure the Materials & Methods section as outlined in our guide to authors, including a completed Reagents and Tools Table that can be downloaded from our author guidelines as well (<https://www.embopress.org/page/journal/14602075/authorguide#structuredmethods>).

10) Digital image enhancement is acceptable practice, as long as it accurately represents the original data and conforms to community standards. If a figure has been subjected to significant electronic manipulation, this must be clearly noted in the figure legend and/or the 'Materials and Methods' section. The editors reserve the right to request original versions of figures and the original images that were used to assemble the figure. Finally, we generally encourage uploading of numerical as well as gel/blot image source data; for details see: embopress.org/page/journal/14602075/authorguide#sourcedata

At EMBO Press, we ask authors to provide source data for the main manuscript figures. Our source data coordinator will contact you to discuss which figure panels we would need source data for and will also provide you with helpful tips on how to upload and organize the files.

In the interest of ensuring the conceptual advance provided by the work, we recommend submitting a revision within 3 months (21st May 2025). Please discuss the revision progress ahead of this time with the editor if you require more time to complete the revisions. Use the link below to submit your revision:

Link Not Available

Rev_Com_number: RC-2024-02794
New_manu_number: EMBOJ-2025-120456-T
Corr_author: Morris
Title: The SUMO activating enzyme subunit, SAE2, contributes SUMO protein bias for mitotic fidelity.

Response to Reviewers

Reviewer #1 (Evidence, reproducibility and clarity (Required)):

Summary:

In their manuscript, Walker et al. investigate the physiological role of deacetylation of the SAE2 subunit of the SUMO E1 enzyme. They find that SAE1:SAE2-acK164 is deacetylated in an HDAC6-dependend manner and use a series of biochemical assays to show that deacetylation of the SAE2 subunit shifts the bias of the SUMO E1 towards SUMO1 conjugation in vitro, proposing a mechanism that is similar to the one that the NEDD8 E1 employs to discriminate between NEDD8 and ubiquitin.

The authors continue to examine the role of different SAE2 variants in different cellular stresses and show that the acetyl-mimicking SAE2K164Q variant displays reduced levels of high molecular weight SUMO1 conjugates in mitotic cells. This variant cannot support proper mitotic spindle formation leading to the appearance of multipolar spindles and centromere-containing micronuclei. Finally, they go on to identify the mechanism underlying these phenotypes and examine NuMA SUMOylation. They test SUMOylation-refractive NuMA variants as well as an already published SUMO1-NuMA fusion that mimics the SUMOylated protein form. They propose a model in which deacetylation of SAE2 changes the bias of the SUMO E1 to increase SUMO1-NuMA conjugation during mitosis, promoting bipolar spindle formation.

Major point:

As the authors state, SUMO1 conjugates decrease during mitosis and this is somewhat at odds with the proposed model regarding NuMA. The authors can detect a SUMOylated NuMA conjugate (fig. 4a). To test whether the proposed model is correct, the authors could check:

- a. Whether this form is indeed SUMO1-NuMA
- b. Whether it decreases upon expression of the SAE2K164Q variant.

These are important points. To answer them, we have used a GFP-tagged NuMA fragment (previously shown to recapitulate NuMA functions in mitosis). We find SUMO1 modification in mitosis that is dependent on HDAC6 activity and suppressed by K164Q-SAE2 (Figure 4a).

Minor points:

1. Fig. 2c: Why does C173G form a thioester with SUMO2 up to 40% of the WT?
2. Please clarify the use of Dox addition in the text and legend earlier (is found currently in Supp. Fig 4).
3. Fig. 4f: what is the difference between the first (invisible NUMA) bipolar and the second, NuMA visible bipolar spindle?
4. ML972- should read ML792 on pg 8.

1. We re-examined the fluorescently tagged SUMO SDS-PAGE gels from which this data is derived, noting that the measurement was affected by a high level of background (evident in Fig. EV3e). We re-analysed the data – now shown in Fig. 2C, by taking intensity readings of

the 120 kDa SAE2-SUMO bands and 'blank' readings immediately below each respective 120 kDa band for subtraction. Doing so has eliminated the impact of variable background between the lanes of the gel.

2. Inserted before first result using doxycycline: "Furthermore, we generated U2OS with a doxycycline-inducible (wild-type) WT FLAG-SAE2 or a FLAG-SAE2-K164R mutant."

3. Fig. 4f now annotated with 'Untransfected' and 'GFP-NuMA transfected'.

4. Corrected, many thanks.

Reviewer #1 (Significance (Required)):

General assessment:

This is a thorough study with complex but well controlled experiments and contains a large amount of valuable information. A point could be further clarified in order to provide further support the proposed model.

Advance:

The document brings understanding on the regulation of the SUMO conjugation system a step forward and links it to a physiological context.

Audience:

basic science: the Ubiquitin family field and also the mitosis-cytoskeleton field. Applied science concerning the use of SUMO inhibitors in cancer.

Expertise: SUMO regulation of the cytoskeleton during mitosis (yeast system)

Reviewer #2 (Evidence, reproducibility and clarity (Required)):

Summary:

Walker et al characterized lysine 164 acetylation of the catalytic SUMO activating enzyme subunit SAE2 and observed that this modification causes a bias towards SUMO2/3 over SUMO1 involving their C-terminal tails. While several enzymes appear to mediate SAE2 acetylation, HDAC6 is responsible for deacetylating SAE2 in mitosis, thereby promoting mitotic SUMO1 modification. The nuclear mitotic apparatus, NuMA, was identified as a putative mitotic SUMO1 substrate upon SAE2 deacetylation. Replacement of endogenous SAE2 with an acetylation mimetic SAE2-K164Q mutant restricts SUMO1 conjugation of NuMA resulting in multipolar spindle formation that can be rescued either by overexpression of SUMO1 or by SUMO1-NuMA fusion.

Major comments:

- Figures 2 C/Supplementary Figure 3c: The enzyme concentrations used in these reactions are much too high. To discriminate between thioester- and isopeptide-linked SUMO, the same samples should be analyzed in the absence (detection of thioester and isopeptide linkages) and presence of high concentrations of DTT (detection of isopeptide-linked SUMO only). The presented assay is problematic as it shows dimeric SUMO and RanGAP1:SUMO bands in the absence of ATP and no UBC9 but SAE2 thioester/isopeptide formation in the

absence of RanGAP1 (preferentially UBC9 should form a thioester/isopeptide bond in this condition as higher molarities of UBC9 over E1 are used). Dimeric SUMO should not be detected unless disulfide bridges are formed between cysteines - this happens when DTT is not present in the reaction - under such conditions, SAE2 and UBC9 can also form disulfide bridges via their catalytic cysteines, impairing their enzymatic activity. In order to interpret the results correctly, it is important to add low concentrations of DTT (~0.1 mM) even in thioester reactions and to distinguish between thioester and isopeptide linkages.

We have repeated these assays as suggested by the reviewer, reducing the enzyme concentrations and using low-concentration DTT. As the reviewer suggested, this successfully clarified the assay (Fig. EV3 f and g). Many thanks for these critiques.

Further, we have control experiments with and without DTT that demonstrate the sensitivity of the SAE2~SUMO band to the reducing agent (Fig EV3d).

- Figure 2F/ Supplementary Figure 3d: Again, the enzyme concentrations are much too high and need to be reduced to a concentration where mainly RanGAP1 monosumoylation with SUMO1 is detected. As RanGAP1 is the most efficient SUMO substrate known, the enzyme concentrations and reaction time can be greatly reduced to limit the auto-modification of the enzymes and SUMO chain formation. Due to the efficient chain-forming activity of SUMO2, this is more difficult with SUMO2, but can be reduced by limiting the concentration of UBC9 in particular or by using a SUMO2 KallR mutant. In the reaction shown, the authors used only twice the molarity of SUMO compared to the substrate, too low taking into account SUMO2 chain formation, enzyme and substrate modification (The reaction should be limited by enzyme activity not by SUMO2). How can the authors be sure that the band they report as RanGAP1 high MW SUMO2 is indeed RanGAP1 modified and not SAE2 (in comparison to Suppl Figure 3b)?

We repeated these assays as suggested by the reviewer, reducing the enzyme concentrations and using low-concentration DTT. These now produce single products, simplifying the assay and quantification (Fig EV3 f & g).

- Figure 3 nicely shows that ML792-resistant SAE2 variants conjugate SUMO2 equally well, whereas SAE2 K163R is reduced and SAE2 K163Q appears to be abolished in SUMO1 conjugation. However, only high molecular weight SUMO conjugates are shown. What are the levels of free SUMO after overexpression of SAE2 variants and the indicated treatments? According to the work of Zhang et al from the Matunis lab (cited as reference 39 in the proposed study), SUMO conjugation is greatly reduced in nocodazole-arrested cells, but is restored after release in G1. Furthermore, SUMO1 and SUMO2 localize to different subcellular regions during mitosis. Have the authors tested whether SAE2 variants differ in their intracellular localization or alter the subcellular localization of SUMO1 and SUMO2 in interphase and mitotic cells?

We have rerun these experiments using antibodies better able to detect free-SUMOs, and include these data in Fig EV4f.

We have investigated SUMO isoform location. In our hands, using a range of SUMO antibodies, we do not observe the previously reported localisations in wild-type cells; therefore, we are unable to assess the impact of the SAE variants on any SUMO1-3 protein

localisation changes. We have also assessed the localisation of the SAE2 variants (Figure EV4b), noting that they do not differ in localisation from that of the WT enzyme.

Can the authors comment on the proportion of SAE2 that is acetylated?

A comment is included in the discussion.

- Figure 4: The authors show a ML792-sensitive high molecular weight smear of NUMA in nocodazole treated cells. It would be very convincing if the authors could demonstrate whether endogenous NUMA is conjugated to SUMO1 or SUMO2 in mitosis by SUMO IPs and whether they can detect a change upon expression of SAE2 variants as in Figure 3a. By replicating this experiment, it would be important to demonstrate the presence of both free and conjugated SUMO paralogs in the input and paralog specific sumoylation in general (smear) and of NUMA in the IP.

These are important points. To answer them, we have used a GFP-tagged NuMA fragment (previously shown to recapitulate NuMA functions in mitosis). NuMA is a known SUMO1 substrate, and we note that its SUMO1 modification in mitosis is dependent on HDAC6 activity and is suppressed by K164Q-SAE2 (Figure 4a). Free and conjugated SUMO levels are shown. Moreover, we tested the ability of SUMO1 Vs SUMO2 fusion to NuMA to reduce multi-polar spindles in HDAC6i-treated cells, finding that SUMO2- fusion fails to do so (Figure 4f).

Minor comments:

- Supplementary Figure 2: Please indicate the size of the marker bands, the fraction numbers and which fractions were pooled for further analysis. Is there any explanation why SAE1:SAE2K164R eluates in two peaks, suggesting two complexes? How different are they in size?

Ladder markers are now added to each gel image. Fraction numbers added. Black block indicates fractions pooled.

The figure is updated with relevant recombinant protein preps generated for the updated *in vitro* experiments. The additional SAE1:SAE2-K164R peak, which appeared in the previous manuscript Fig. EV2a eluted in the void volume and so we think it comprised aggregated SAE1:SAE2 protein.

Reviewer #2 (Significance (Required)):

The finding that E1 acetylation regulates SUMO paralog specificity is very exciting, particularly because of its link to key regulatory mitotic functions. Overall, the findings are intriguing and supported in part by various biological and biochemical methods. However, some concerns remain unsatisfactorily addressed, as outlined above.

The findings provide a novel basic concept of how E1 enzyme regulation contributes to the specification of modifier selectivity, demonstrates cross-talk with other PTMs and reveals a biological function. Therefore, the study is of interest to a broad audience.

Reviewer #3 (Evidence, reproducibility and clarity (Required)):

Summary:

In this manuscript, the authors report on an interesting regulatory mechanism that influences the balance between conjugation of the different SUMO isoforms, SUMO1 versus SUMO2/3. The authors describe that acetylation of a specific residue, K164, in the SUMO activating enzyme (E1) subunit, SAE2, biases the E1's preference towards SUMO2/3. Specifically, they use an acetylation-mimicking K164Q mutation to show that the acetylation state of SAE2 likely affects the affinity of the E1 to SUMO and the rate of thioester formation. With an antibody, they demonstrate the acetylation of SAE2 in cells. Mechanistically, they locate the cause of the isoform bias to a residue in the C-terminus of SUMO in proximity to K164 or SAE2, where SUMO1 carries glutamate, while SUMO2/3 has glutamine. Switching these residues between the SUMO isoforms reverses the isoform preference of the E1. Phenotypically, the SAE2 K164Q mutant induces mitotic problems that the authors attribute to the SUMOylation of the NuMA complex. They assign the deacetylation of SAE1 to HDAC6 and report that deacetylation occurs during mitosis. These results are consistent with a model that SUMO1 modification of the NuMA complex in mitosis is important for mitotic fidelity and that the cell cycle-dependent changes in the acetylation status of SAE2 promote this. Accordingly, fusion of SUMO1 to a NuMA subunit partially overcomes the problems induced by the K164Q mutant or the inhibition of HDAC6.

Major comments:

The experiments are largely performed in a well-controlled manner, and overall, the study is very convincing. I would like to suggest a few experiments that would strengthen the authors' conclusions, and there are a few minor issues with some of the figures.

1. It would be helpful if the authors could more clearly separate the two steps catalyzed by the E1. This would be needed to determine whether the accumulation of the SUMO1-AMP intermediate by the K164Q mutant is due to a faster rate of formation or a reduced rate of conversion to the thioester. They could test the AMP formation step in isolation in a straightforward manner by using the double mutant K164Q C173G and measuring a time course of SUMO1-AMP versus SUMO2-AMP build-up. Alternatively, they could try to isolate the second step by adding SUMO1-AMP versus SUMO2-AMP to the E1 de novo - although isolation of the intermediates may be more involved.

We implemented the first approach suggested, creating a new double mutant, C173G-K164Q, and compared the rates of SUMO1-AMP-BODIPY generation with those of C173G. Using this assay, we are unable to detect a difference between. While this finding might suggest the K164Q is no faster than C173G, we are reluctant to include the data in the manuscript. This is because we found the SUMO1-AMP-BODIPY generation was quick, saturated after 30 seconds in an experiment performed at 4 oC, so that our sampling error is likely large, and very difficult to discount in this experimental set up.

2. The reason for the isoform selectivity in the context of NuMA SUMOylation remains unresolved. The study would be significantly strengthened if the authors could address the question of whether the mitotic defects come from a lack of NuMA SUMOylation or the wrong type of SUMOylation. In other words, does it matter which isoform of SUMO is attached to NuMA? This could be addressed by also creating a SUMO2 fusion construct and testing whether that suppresses some of the phenotypes observed with the K164Q mutant and upon HDAC6 inhibition.

This is an excellent suggestion. We tested the idea by making and testing the SUMO2-fusion as suggested. In contrast to the SUMO1-NuMA fusion, the SUMO2-NuMA did not suppress

multipolar spindles (Fig 4f). This finding suggests that SUMO1 modification, rather than modification itself, is important.

3. The authors should include a more detailed discussion of the importance of the absolute and relative concentrations of free SUMO1 versus SUMO2/3 as a possible mechanism to impose isoform bias. Specifically, they should consider the different KM values of the E1 for the isoforms. The literature says that the E1 has a lower KM (higher affinity) for SUMO1 than SUMO2/3 but also a lower kcat (considering both steps of its reaction together), resulting in an approximately equal Kcat/KM. This would mean that at low overall SUMO concentrations, SUMO1 would have an advantage, whereas with rising SUMO concentrations SUMO2/3 would be favoured (which might be particularly important during stress conditions). What part of the curve does the cellular environment reflect?

Yes, this is a good suggestion, we have included this in the discussion.

4. It would be helpful to show a time course of endogenous SAE2 acetylation over the cell cycle, using synchronized cultures. All the experiments showing acetylation are done with transfected FLAG-tagged constructs - are they overexpressed?

The FLAG-tagged constructs are not over-expressed relative to endogenous levels of SAE2, with the exception of the C173G variant, which is. Figure EV4a makes this clear.

We have attempted to examine endogenous acetylated SAE2 using immunoprecipitated endogenous SAE2 protein, but our results are not of a quality we wish to report. Our experiments required 3 x 15 cm dishes of cells for SAE2 immunoprecipitation with the available antibody. Blotting for the acK164 revealed a very faint band. Any further increase in lysate concentration introduced greater noise, reducing our ability to analyse the acK164. We are reluctant to claim anything about this, not knowing whether the SAE2-immunoprecipitation precludes detection of the modification or how sensitive our anti-acK164 antibody is. We include a statement in the discussion that highlights the limitations of our study.

We have been more successful assessing the modification of FLAG-SAE2 (using anti-FLAG to precipitate) at various times and treatments through the cell cycle, which is included here for the reviewer's interest.

Minor comments:

- The title is not immediately understandable. "SUMO protein bias for mitotic stability" sounds a bit awkward. It would be clearer to be more explicit about isoforms.

We have changed the title to be more explicit:

"HDAC6-Dependent Deacetylation of SAE2 enhances SUMO1 Conjugation for Mitotic Integrity"

- On page 3, the authors could introduce a justification of why they tested IR treatment.

Done

- The authors repeatedly use the word "codon" when they describe a site in the protein. Codon refers to mRNA, so the word "residue" would be more appropriate when talking about a protein.

Agreed. Done.

- Page 8: "confirmation" should be "conformation". Done

- Page 8: "While we find a little role for..." - delete "a" Done

- Fig 2a: The figure would be easier to understand if the same colour scheme was used for S1 versus S2 to aid the comparison. Changed.

- Fig 2b: I don't understand the units of this graph. Why does normalization result in a value of zero, not 1? On this scale, what would a value of 1 signify? How can a value become negative? I would have expected values relative to the WT, with the WT being set to 1 or to 100%. The authors should also show the raw data for this plot.

It is normalised to WT condition which =1 instead of 0, and raw representative data is shown in Fig. EV3.

- Fig 2c: Please also show representative raw data. Representative images for SAE2~SUMO1-Alexa488 and SAE2~SUMO2-Alexa647 and sypro stain for SAE1:SAE2 loading is shown Fig EV3.

- Fig 2d,f: Again, the legend should explain what the plots were normalized to. Inserted in the legend for Fig. 2d&f:

"The RanGAP1-SUMO1 products are normalised to the WT SAE1:SAE2:SUMO1-only condition (top) and the RanGAP1-SUMO2 products are normalised to the WT SAE1:SAE2:SUMO2-only condition (bottom)."

- Fig 3g: Could the authors comment on the detrimental effects of both SUMO1 and SUMO2 in the WT background? Comment included.

- Fig 3h: typo ("Trasfect") Done

- Fig 4b: The images have very poor contrast. In addition, the merged image would be clearer if two different colours were used. Improved.

- Fig 4f: The DAPI signal is hardly visible - better contrast would help. Improved.

- Fig S2: It would be appropriate to indicate which fractions were actually collected or combined during the purification.

Ladder markers have been added to each gel image. Fraction numbers added. The black box indicates the fractions pooled.

- Fig S5b: The authors argue with the hydrogen bonding capacities of the different pairings. However, acetylation at K164 should not necessarily prevent a hydrogen bond to SUMO1-

E93, considering that the "NH" group is likely still at a comparable distance to the carboxylate of E93 and could in principle undergo H-bonding unless prevented by the steric bulk introduced by the acetyl group. On the other hand, the K164-E93 interaction is the only electrostatic interaction among the 4 possible combinations. While a contribution is not easy to prove experimentally, I think the possibility of charge-charge interactions having an impact should be considered in the discussion.

Agreed, it is possible that altered hydrogen bonding, or the electrostatic interactions of K164:E93, contribute to the SUMO1/2 Vs SUMO1 bias of the modified and unmodified enzyme. This is now included.

Reviewer #3 (Significance (Required)):

Significance:

The results presented here are interesting and novel. Importantly, the authors provide a molecular model for a new mechanism of how the SUMO system achieves isoform specificity, which is a still very poorly understood phenomenon. The manuscript makes a significant advance by contributing an important new aspect of how the SUMO conjugation machinery chooses between isoforms. The manuscript is strong by providing very good evidence for its conclusions. One limitation is the strong reliance on the use of an acetyl-mimicking mutant; this limitation could be overcome by placing a bit more emphasis on detecting endogenous SAE2 acetylation.

Audience: The study should be relevant to a broad audience, given the impact of the SUMO system on cellular regulation; after all, the study addresses a very fundamental problem in the field. In addition, it should be of interest to researchers studying regulation of mitosis.

Response: We have complemented our assays that used FLAG-tagged constructs with observations using HDAC6-inhibitor, the deacetylase largely responsible for the removal of acK164. We find that its inhibition phenocopies the impact of the K164R variants without the need to express an exogenous protein. This approach mitigates some of the concern that conclusions are based solely on an exogenous protein mimic, although we acknowledge that the inability to detect endogenous SAE2 acetylation is a limitation, which is discussed in the manuscript and highlighted for readers.

Prof. Joanna R Morris
University of Birmingham
School of Medical Sciences
College of Medicine and Health
Birmingham, West Midlands B15 2TT
United Kingdom

14th Jul 2025

Re: EMBOJ-2025-120456R
HDAC6-Dependent Deacetylation of SAE2 enhances SUMO1 Conjugation for Mitotic Integrity.

Dear Jo,

Thank you for submitting your revised manuscript to The EMBO Journal. It has now been seen again by two of the original Review Commons referees, and I am pleased to say that they were broadly satisfied with your responses and revisions. Both still note a few presentational issues (see below), which I would ask you to address during a final round of minor revision. In addition, there are the following editorial items remaining:

- Please carefully go through the reference list and make sure that each reference is complete with citation year, volume, and page/locator numbers (which are currently lacking for several of them).
- Please provide suggestions for a short 'blurb' text prefacing and summing up the study in two sentences (max. 250 characters), followed by 3-5 one-sentence 'bullet points' with brief factual statements of key results of the paper; they will form the basis of an editor-written 'Synopsis' accompanying the online version of the article. Please also upload a synopsis image, which can be used as a "visual title" for the synopsis section of your paper (maybe based on a slightly compacted/rearranged version of Figure 5?). The image should be in PNG or JPG format with the modest dimensions of EXACTLY 550 pixels wide and 300-600 pixels high.

I am therefore returning the manuscript to you for a final round of revision, to allow you to make these modifications and upload the revised files. Once we will have received them, we should hopefully be able to proceed with formal acceptance and production of the manuscript.

With kind regards,

Hartmut

- a point-by-point response to the referees' comments, with a detailed description of the changes made (as a word file).
- a word file of the manuscript text.
- individual production quality figure files (one file per figure)
- a complete author checklist, which you can download from our author guidelines (<https://www.embopress.org/page/journal/14602075/authorguide>).

- Expanded View files (replacing Supplementary Information)

- a Reagents and Tools Table as part of the Methods section, which can be downloaded from our author guidelines

(<https://www.embopress.org/page/journal/14602075/authorguide#structuredmethods>)

Revision to The EMBO Journal should be submitted online within 90 days, unless an extension has been requested and approved by the editor; please click on the link below to submit the revision online before 12th Oct 2025:

Link Not Available

Referee #1:

The revised version of the manuscript by Walker et al has significantly improved and the authors have satisfactorily addressed all my concerns. New insights into how SUMO enzymes discriminate between SUMO paralogs, and their biological consequences, are intriguing and of great interest to a wider community. The findings are supported by various biological and biochemical methods. Together, these findings warrant publication in The EMBO Journal.

However, the manuscript still has several minor flaws that need correcting. For example, the last two paragraphs of the introduction are repeated twice on pages 2 and 3, and there are several incomplete sentences, usually after page breaks, as well as misalignments in the figures, which may be formatting issues.

e.g. page 5

SUMO E1 regulation directs variant bias for mitotic fidelity.
contrast, when incubated with...»

page 6

SUMO E1 regulation directs variant bias for mitotic fidelity.
prometaphase with nocodazole

page7

SUMO E1 regulation directs variant bias for mitotic fidelity.
depleted or complemented cells

etc

also some Figures show misalignments

Fig. 1B, Figure 4 multiple subfigures.

Referee #2:

The authors have done a very good job in addressing the reviewers' comments. Their new experiments have significantly strengthened the manuscript and have further enhanced my enthusiasm for this study. Not all of the experiments they performed in response to my previous queries have worked out, but the authors discuss the resulting limitations transparently. Overall, I therefore support publication in the EMBO Journal. There are a few minor issues left that would need attention before publication:

1. pg 4-5: It would be very helpful if the authors included some measure of quantification in their discussion of the SUMO1-versus SUMO2-bias. Currently, we read that there is a bias, but it requires very careful scrutiny of the figure to judge the extent of this bias.

2, pg 9: The possibility of charged interactions versus H-bonding being responsible for the observed effects should be included in the discussion of Fig. EV5B.

3. pg 9: There is a statement that is broken off mid-sentence: "Indeed, our experiments required."

4. Figures: Please go over all figures again and clean up the labeling of graphs and arrangement/alignment of boxes, labels and other text. At present, there are multiple instances of bars or lines crossing labels, scattered labels (e.g., "nr"), varying font sizes, varying sizes of microscopic images, and poor alignments.

Rev_Com_number: RC-2024-02794
New_manu_number: EMBOJ-2025-120456R
Corr_author: Morris
Title: HDAC6-Dependent Deacetylation of SAE2 enhances SUMO1 Conjugation for Mitotic Integrity.

Responses to editorial requests and reviewers' comments (EMBOJ-2025-120456R )

Editorial requests.

-Please provide suggestions for a short 'blurb' text prefacing and summing up the study in two sentences (max. 250 characters), followed by 3-5 one-sentence 'bullet points' with brief factual statements of key results of the paper; they will form the basis of an editor-written 'Synopsis' accompanying the online version of the article. Please also upload a synopsis image, which can be used as a "visual title" for the synopsis section of your paper (maybe based on a slightly compacted/rearranged version of Figure 5?). The image should be in PNG or JPG format with the modest dimensions of EXACTLY 550 pixels wide and 300-600 pixels high.

Response:

The image is uploaded. Below is the suggested text for 'blurb'.

SAE2-K164 deacetylation by HDAC6 during mitosis promotes SUMO1 conjugation, which is essential for proper spindle formation and mitotic fidelity. An acetyl-mimetic at SAE2-K164 skews SUMO variant usage and results in chromosome segregation errors.

- SAE2 is deacetylated at lysine-164 in an HDAC6-dependent manner during mitosis.
- Acetyl-mimetic SAE2-K164Q squanders the SUMO1-adenylate intermediate, and biases SUMO2 over SUMO1 conjugation.
- SAE2-K164Q expression suppresses mitotic SUMO1ylation and induces multipolar spindles.
- SUMO1 overexpression or a SUMO1-NuMA fusion rescues mitotic defects from SAE2-K164Q complementation or HDAC6 inhibition.
- SUMO1, but not SUMO2, conjugation to NuMA supports bipolar spindle formation.

- Please carefully go through the reference list and make sure that each reference is complete with citation year, volume, and page/locator numbers (which are currently lacking for several of them).

Response: The references lacking page numbers/identifiers have been added.

Reviewers comments:

Referee #1:

The revised version of the manuscript by Walker et al has significantly improved and the authors have satisfactorily addressed all my concerns. New insights into how SUMO enzymes discriminate between SUMO paralogs, and their biological consequences, are intriguing and of great interest to a wider community. The findings are supported by various biological and biochemical methods. Together, these findings warrant publication in The EMBO Journal.

However, the manuscript still has several minor flaws that need correcting. For example, the

last two paragraphs of the introduction are repeated twice on pages 2 and 3, and there are several incomplete sentences, usually after page breaks, as well as misalignments in the figures, which may be formatting issues.

e.g. page 5

SUMO E1 regulation directs variant bias for mitotic fidelity.

contrast, when incubated with...»

page 6

SUMO E1 regulation directs variant bias for mitotic fidelity.

prometaphase with nocodazole

page7

SUMO E1 regulation directs variant bias for mitotic fidelity.

depleted or complemented cells

etc

also some Figures show misalignments

Fig. 1B, Figure 4 multiple subfigures.

Response: The duplicated paragraph has been removed. Most of the misalignments (with some exceptions) and all of the incomplete sentences, far as we can tell, were errors produced in the upload process (not part of the files we have). Hopefully, the new uploads will be ok.

Referee #2:

The authors have done a very good job in addressing the reviewers' comments. Their new experiments have significantly strengthened the manuscript and have further enhanced my enthusiasm for this study. Not all of the experiments they performed in response to my previous queries have worked out, but the authors discuss the resulting limitations transparently. Overall, I therefore support publication in the EMBO Journal. There are a few minor issues left that would need attention before publication:

1. pg 4-5: It would be very helpful if the authors included some measure of quantification in their discussion of the SUMO1- versus SUMO2-bias. Currently, we read that there is a bias, but it requires very careful scrutiny of the figure to judge the extent of this bias.

Response: Many thanks for the suggestion – a more quantitative discussion of the bias is now included (bottom of page 4).

2, pg 9: The possibility of charged interactions versus H-bonding being responsible for the observed effects should be included in the discussion of Fig. EV5B.

Response: Now included in the discussion.

3. pg 9: There is a statement that is broken off mid-sentence: "Indeed, our experiments required."

Response: Deleted.

4. Figures: Please go over all figures again and clean up the labeling of graphs and arrangement/alignment of boxes, labels and other text. At present, there are multiple instances of bars or lines crossing labels, scattered labels (e.g., "nr"), varying font sizes, varying sizes of microscopic images, and poor alignments.

Response: These problems are related to the upload.